# ENSEMBLE PREDICTION OF TASK AFFINITY FOR EFFICIENT MULTI-TASK LEARNING

**Afiya Ayman**
Pennsylvania State University

**Ayan Mukhopadhyay**
College of William & Mary

**Aron Laszka**
Pennsylvania State University

## ABSTRACT

A fundamental problem in multi-task learning (MTL) is identifying groups of tasks that should be learned together. Since training MTL models for all possible combinations of tasks is prohibitively expensive for large task sets, a crucial component of efficient and effective task grouping is predicting whether a group of tasks would benefit from learning together, measured as per-task performance gain over single-task learning. In this paper, we propose **ETAP** (Ensemble Task Affinity Predictor), a scalable framework that integrates principled and data-driven estimators to predict MTL performance gains. First, we consider the gradient-based updates of shared parameters in an MTL model to measure the affinity between a pair of tasks as the similarity between the parameter updates based on these tasks. This linear estimator, which we call affinity score, naturally extends to estimating affinity within a group of tasks. Second, to refine these estimates, we train predictors that apply non-linear transformations and correct residual errors, capturing complex and non-linear task relationships. We train these predictors on a limited number of task groups for which we obtain ground-truth gain values via multi-task learning for each group. We demonstrate on benchmark datasets that **ETAP** improves MTL gain prediction and enables more effective task grouping, outperforming state-of-the-art baselines across diverse application domains.

## 1 INTRODUCTION

Multi-task learning (MTL) is an approach to learn multiple machine-learning tasks together to improve generalization performance and reduce inference time Caruana (1997); Standley et al. (2020). MTL improves the performance of individual tasks by leveraging related information in other tasks through *inductive transfer*, i.e., the learning architecture is designed to introduce inductive bias favoring hypothesis classes that perform well across tasks Caruana (1997). MTL has proven to be effective in multiple domains, including computer vision Zhang et al. (2014); Liu et al. (2015a), health informatics Puniyani et al. (2010); Zhou et al. (2011), and natural language processing Luong et al. (2015). However, simply learning a set of tasks together does not ensure better performance; in fact, performance can degrade compared to performing single-task learning (STL) for each task, a phenomenon known as *negative transfer* Rosenstein et al. (2005). This observation highlights a critical challenge: finding groups of tasks that maximize the performance gain from MTL.

A naïve strategy to find well-performing task groups is to run multi-task learning for every possible group; however, this is computationally infeasible in most cases. So, recent research focused on *predicting MTL gains*, that is, predicting the improvement in performance when a group of tasks are trained together versus independently. Accurate prediction of MTL gains enables finding optimal (or near optimal) task groups, leading to more efficient and effective MTL.

A common approach for predicting MTL gains—which we refer to as the *data-driven* or *black-box approach*—is to first obtain ground-truth MTL gain values by performing multi-task learning for several task groups and measuring performance gains, and to then predict gains for other groups based on these ground-truth values. For example, *Higher-Order Affinities* (HOA) relies on ground-truth MTL gains for all task pairs (Standley et al., 2020). Other methods such as MTGNet (Song et al., 2022) and Linear Surrogate (Li et al., 2023) learn meta-models from large numbers of sampled training groups, relying on ground-truth MTL gains from extensive multi-task learning. Despite methodological differences, these approaches share a key limitation: they are computationally ex-

pensive and often require large numbers of training groups to provide accurate, robust predictions. This cost–robustness tradeoff becomes prohibitive as the number of tasks grows.

In contrast, *white-box approaches* aim to predict MTL performance by analyzing training dynamics—specifically, how gradient updates from one task affect the losses of other tasks. A representative example is Task-Affinity Grouping (TAG), which estimates inter-task affinity by training a single MTL model while performing additional hypothetical training steps to evaluate the impacts of task-specific parameter updates (Fifty et al., 2021). While theoretically grounded, TAG is limited to only pairwise estimates, which are highly correlated to MTL gains but are not a direct predictors. TAG also struggles to model higher-order interactions, limiting its accuracy in realistic multi-task settings. Further, it incurs significant computational overhead due to the additional forward/backward passes, making its overall cost comparable to training many separate MTL models.

**Contributions:**   This paper introduces **ETAP** (**E**nsemble **T**ask-**A**ffinity **P**rediction), a novel framework that combines the theoretical strengths of white-box affinity estimation with the generalization ability of data-driven models to predict MTL gains, which is formally defined as the reduction in loss due to learning a group of tasks jointly versus independently. ETAP addresses the limitations of prior approaches for prediction through a two-part integration. First, we propose an improved white-box technique to compute ***task-affinity scores*** from gradient interactions during a single MTL training with minimal computational overhead. Compared to prior methods such as TAG, our approach is more efficient and avoids auxiliary training steps, while producing reliable affinity estimates. However, leveraging these affinity scores to accurately predict higher-order MTL gains poses a challenge: task dependencies are often non-linear and cannot be fully captured by pairwise relationships alone. To address this, ETAP employs a ***two-stage ensemble prediction framework***. The first stage learns a non-linear mapping from affinity scores to predicted group-level MTL gains. The second stage uses a residual model to correct systematic prediction errors by learning from mismatches between predictions and ground-truth gains. This design improves robustness and enables accurate predictions with far fewer training groups than prior data-driven methods.

Our MTL-gain predictor may be integrated as an objective into a wide range of search algorithms to find optimal task groups for MTL (e.g., it can be integrated into branch-and-bound algorithms from Standley et al. (2020) and Fifty et al. (2021) or beam-search algorithm from Song et al. (2022)). In our experimental evaluation, we use the efficient branch-and-bound search algorithm from Standley et al. (2020) to find task groups with our proposed predictor and with baseline predictors. Our experimental results demonstrate that ETAP outperforms existing data-driven and white-box methods in terms of both accuracy and computational efficiency across diverse MTL benchmarks. By strategically integrating white-box affinity measurements with data-driven refinement, ETAP provides a better tradeoff between efficiency and accuracy, addressing the limitations of existing approaches.

## 2   PROBLEM DEFINITION

We consider a set of $n$ tasks, denoted $\mathcal{T} = \{t_1, t_2, \cdots, t_n\}$, $n = |\mathcal{T}|$. A task group, denoted $G$, is a subset of $\mathcal{T}$ (i.e., $G \subseteq \mathcal{T}$) and can have any number of tasks from $1$ to $n$. An arbitrary set of such groups is denoted as $\mathcal{G}$ (i.e., $\mathcal{G} \subseteq 2^{\mathcal{T}}$).

A **multi-task learning (MTL)** algorithm jointly trains on the data of all tasks in a group $G$ to build a shared feature representation. In many architectures, task-specific decoders transform this shared representation to produce task-specific predictions Ruder (2017); Crawshaw (2020). The MTL algorithm trains a multi-task model for tasks in group $G$ with parameters $\theta_{\text{MTL}}^G$. The loss for a specific task $t$, denoted by $L_t$, measures the model's prediction performance on that task. During training, each task updates the model parameters $\theta_{\text{MTL}}^G$ based on its own loss function.

**MTL gain** is our formal metric for evaluating task relationships in the context of MTL performance. MTL gain measures the performance improvement for a task (i.e., reduction in loss $L_t$) when trained using MTL compared to training the task independently using a single-task learning (STL) model. Specifically, for a task $t \in G$, the relative MTL gain, denoted by $y_{G \rightarrow t}$, is the reduction in loss for task $t$ when tasks in group $G$ are trained together. Formally, the relative MTL gain for task $t \in G$ is

$$y_{G \rightarrow t} = \frac{L_t(\mathcal{X}, \theta_{\text{STL}}^t) - L_t(\mathcal{X}, \theta_{\text{MTL}}^G)}{L_t(\mathcal{X}, \theta_{\text{STL}}^t)}. \tag{1}$$

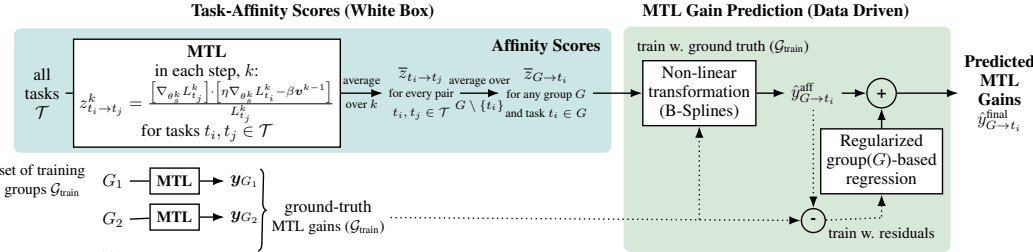

Figure 1: Visualizing ETAP: From White-box Task Affinity Scores to Data-driven Ensembled MTL Gain Predictions. Task affinity computed from a baseline MTL model and ground-truth MTL gains are fed into an ensemble framework. Non-linear transformations yield initial predictions, which are later refined by residual correction through regularized regression.

where $\mathcal{X}$ is the test dataset, $L_t(\mathcal{X}, \theta_{\text{STL}}^t)$ is the loss of task $t$ when trained independently to obtain parameters $\theta_{\text{STL}}^t$, and $L_t(\mathcal{X}, \theta_{\text{MTL}}^G)$ is the loss of task $t$ when trained using MTL with group $G$ to obtain parameters $\theta_{\text{MTL}}^G$. A positive value of $y_{G \to t}$ indicates a performance gain (i.e., loss reduction), while a negative value indicates performance degradation.

Given task-set $\mathcal{T}$, the goal is to identify **optimal task groups that maximize MTL performance**. This involves minimizing the total loss $L_{\mathcal{T}}$ across all tasks in $\mathcal{T}$ or, equivalently, maximizing the overall MTL gains for each task. Formally, we aim to select a collection of task groups $\mathcal{G} = \{G_1, G_2, \dots\}$, under a constraint on the number of groups $|\mathcal{G}| \leq B$, to maximize the sum of MTL gains for each task across all groups:

$$\text{Maximize:} \quad \sum_{t \in \mathcal{T}} \max_{G \subseteq \mathcal{T}} y_{G \to t}, \quad \text{subject to} \quad |\mathcal{G}| \leq B. \tag{2}$$

This optimization problem is NP-hard due to the exponential number of possible task groups. Since an exhaustive search is infeasible, we focus on efficiently predicting MTL gains for task groups. The prediction of MTL gains involves learning a function $f$ that estimates $\hat{y}_{G \to t}$ for a task $t \in G$ by minimizing prediction error $(\hat{y}_{G \to t} - y_{G \to t})^2$. Our goal is to minimize both the error in MTL gain predictions as well as the computational cost of learning $f$, enabling efficient and effective task grouping for improved MTL performance.

## 3  ETAP: ENSEMBLE TASK-AFFINITY PREDICTION

Our proposed approach, **ETAP**, is a unified framework for predicting MTL gains. ETAP predicts which groups of tasks will benefit from joint training (benefit measured as performance gain over STL) and use these predictions to form effective task groups, avoiding exhaustive search.

ETAP combines principled white-box analysis with data-driven modeling. It first computes gradient-based *task affinity scores* by analyzing how updates from one task affect another's loss, providing model-aware estimates of linear relationships. These scores form the foundation of gain prediction. Next, ETAP employs data-driven models to refine these estimates, learning the complex, non-linear relationships between predicted and actual MTL gains. By integrating these complementary methods, ETAP achieves accurate gain prediction with limited supervision. Finally, using the predictions made by ETAP, we select a near-optimal set of task groups that maximizes total MTL gain while balancing task coverage and performance under a budget constraint on the number of groups.

### 3.1  TASK-AFFINITY SCORE VIA WHITE-BOX ANALYSIS

Our goal is to efficiently estimate the affinity between pairs of tasks, capturing how their performance improves when trained together using multi-task learning. During training, tasks implicitly exchange information through successive gradient updates to the shared parameters. We collect task-specific losses, gradients, and parameter updates at each training step to compute pairwise task-affinity scores. These affinity scores correlate well with ground-truth MTL gains (as demonstrated in Section 4.2) and serve as a robust and computationally inexpensive baseline predictor.

### 3.1.1 TASK LOSS AND GRADIENTS

We begin by training a single MTL model for a group with all tasks in $\mathcal{T}$. The MTL model has a shared encoder with parameters $\theta_s$ ($_s$ for shared) and task-specific decoders for each task where parameters for task $t_i$ are represented as $\theta_{t_i}$. In gradient step $k$, for a batch of examples $\mathcal{X}^k$, the loss for task $t_i$ is $L_{t_i}^k = L_{t_i}\left(\mathcal{X}^k, \theta_s^k, \theta_{t_i}^k\right)$. In each step, we compute the gradient of each task's loss $L_{t_i}^k$ with respect to the shared parameters, $\theta_s$, which measures how a small change in $\theta_s$ affects task loss $L_{t_i}$ using the following:

$$\nabla_{\theta_s^k} L_{t_i}^k = \nabla_{\theta_s^k} L_{t_i}\left(\mathcal{X}^k, \theta_s^k, \theta_{t_i}^k\right) \tag{3}$$

These gradients encode the relationship between shared parameters and task losses, capturing task-specific sensitivities to changes in $\theta_s$. Note that these gradients are already computed as part of the standard backpropagation process during MTL training, so no additional computation is required.

### 3.1.2 SHARED PARAMETER UPDATES

Our approach assumes a standard MTL setup with shared encoder parameters updated via gradient descent, optionally with momentum. While not specific to our method, we leverage this common MTL optimization behavior to quantify task interactions. Understanding how tasks interact through updates on shared parameters is critical, as these updates reveal how tasks impact each other's performance, providing key insights for estimating task affinities. Let $\theta_{s|t_i}^{k+1}$ denote the updated shared parameters $\theta_s$ *after* gradient-step $k$ taken using only task $t_i$'s loss. If momentum is used during training, the update incorporates previous step's velocity $\boldsymbol{v}^{k-1}$ and momentum coefficient $\beta$ Sutskever et al. (2013). The corresponding parameter update $\Delta\theta_{s|t_i}^{k+1}$ with learning rate $\eta$ is:

$$\Delta\theta_{s|t_i}^{k+1} = \beta\boldsymbol{v}^{k-1} - \eta\nabla_{\theta_s^k} L_{t_i}^k \tag{4}$$

The velocity term is updated as $\boldsymbol{v}^k = \beta\boldsymbol{v}^{k-1} - \eta\nabla_{\theta_s^k} L_{t_i}^k$. Without momentum, Equation (4) becomes $\Delta\theta_{s|t_i}^{k+1} = -\eta\nabla_{\theta_s^k} L_{t_i}^k$

### 3.1.3 AFFINITY SCORE

We calculate a task-affinity score for each training step, which we later aggregate to estimate the overall relationship between pairs of tasks. To calculate the step-level task-affinity score between tasks $t_i$ and $t_j$ at training step $k$, we measure how the gradient update for task $t_i$ affects the loss for task $t_j$. Specifically, we take the alignment between the gradient for task $t_j$ and the parameter updates $\Delta\theta_s$ based on task $t_i$, normalized by task $t_j$'s loss:

$$z_{t_i \to t_j}^k = \frac{\left[\nabla_{\theta_s^k} L_{t_j}^k\right] \cdot \left[\eta\nabla_{\theta_s^k} L_{t_i}^k - \beta\boldsymbol{v}^{k-1}\right]}{L_{t_j}^k} \tag{5}$$

Here, $z_{t_i \to t_j}^k$ is the affinity score from task $t_i$ to task $t_j$ at step $k$. A positive value means that task $t_i$'s update to the shared parameters $\theta_s$ reduces the loss of $t_j$, suggesting a beneficial transfer, while a negative value suggests interference. We aggregate these scores over all $K$ training steps to obtain a stable estimate of pairwise affinity:

$$\overline{z}_{t_i \to t_j} = \frac{1}{K}\sum_{k=1}^{K} z_{t_i \to t_j}^k \tag{6}$$

Time-averaging the affinity score $z_{t_i \to t_j}^k$ over $K$ training steps is crucial to providing a robust estimate of pairwise affinity since the per-step affinity score $z_{t_i \to t_j}^k$ varies significantly during training (see Figure 7 in Section F for an illustration). The time-averaged pairwise task-affinity scores $\overline{z}_{t_i \to t_j}$ provides a very stable estimate, which exhibits low variance between training runs (see Table 2).

After computing pairwise task-affinity scores $\overline{z}_{t_i \to t_j}$, we derive group-level scores for task groups with three or more tasks. For task $t_i \in G$, the affinity score $\overline{z}_{G \to t_i}$ is the average of pairwise affinities from all other tasks in $G$ to $t_i$:

$$\overline{z}_{G \to t_i} = \frac{1}{|G| - 1}\sum_{t_j \in G \setminus \{t_i\}} \overline{z}_{t_j \to t_i}. \tag{7}$$

Here, $\overline{z}_{t_j \to t_i}$ is the pairwise affinity score from task $t_j$ to task $t_i$. We collect these scores for all tasks within a group $G$ into a set $z_G = \{\overline{z}_{G \to t} : t \in G\}$. In the absence of momentum, due to the linearity of gradient-based measurements, averaging pairwise affinity scores across a group is equivalent to calculating the score based on the group's averaged gradients. This enables simplifying the calculation without sacrificing accuracy and serves as an efficient way to handle large task groups.

While these groupwise affinity scores $\overline{z}_{G \to t}$ offer stable, low-variance estimates due to the averaging over task pairs and training steps, they have two fundamental limitations. First, affinity scores reflect gradient correlations, which may *differ in scale from MTL gains*. Second, affinity-based group predictions are effectively *linear approximations*, which tend to have low variance due to averaging but high bias due to oversimplification, overlooking complex, non-linear task interactions.

## 3.2 DATA-DRIVEN ENSEMBLE PREDICTION

To address the limitations of affinity scores, we introduce a two-stage ensemble framework for predicting MTL gains using a combination of white-box affinity scores and data-driven methods. Building on groupwise affinity scores $\overline{z}_{G \to t}$, which provide first-order approximations of a group's potential for positive transfers, we apply (1) an **affinity-to-gain mapping** that is trained to translate affinity scores into predicted MTL gains, and (2) a **residual correction** that refines these predictions by modeling residual errors between these initial predictions and observed ground-truth MTL gains, further reducing bias and capturing higher-order task interactions. For both stages, we use the same training set of task groups $\mathcal{G}_{\text{train}}$ with measured ground-truth MTL gains. By integrating white-box and data-driven methods, our framework balances the tradeoff between bias and variance, providing accurate predictions of MTL gain with less training data than existing data-driven methods.

### 3.2.1 AFFINITY SCORES TO MTL GAINS WITH NON-LINEAR MAPPING

Since affinity scores differ in scale from MTL gain values, we introduce a non-linear transformation that maps affinity scores $z_G$ to predicted MTL gains $\hat{y}_G^{\text{aff}}$. Besides bridging the gap between affinity scores and MTL gains—two correlated but distinct metrics—the non-linearity of this transformation also enables capturing higher-order task dependencies, which groupwise affinity scores ignore due to their linearity. We learn this transformation from ground-truth MTL gains in training set $\mathcal{G}_{\text{train}}$, using B-spline basis expansion. B-splines are piecewise polynomial functions with local support Boor (1978); Hastie et al. (2009), which we use to map affinity scores to a higher-dimensional feature space. Specifically, for groupwise affinity score $\overline{z}_{G \to t}$ for task $t \in G$, the basis expansion is:

$$\phi\left(\overline{z}_{G \to t}\right) = \left[N_i\left(\overline{z}_{G \to t}\right)\right]_{i=1}^{M} \tag{8}$$

$N_i(\cdot)$ represents the $i$-th B-spline basis function, and $M$ denotes the total number of basis functions, depending on the chosen spline degree and the number of knots. After applying the B-spline expansion, we fit a regularized linear regression model to these higher-dimensional features, balancing model complexity and generalization to control for potential overfitting. Note that while the regression itself is linear, the complete transformation from groupwise affinity scores to predicted MTL gains is non-linear. Formally, the regression model $f_{\text{non-linear}}$ maps affinity scores $z_G$, expanded by B-splines $\phi$, using regression coefficients $\omega_{\text{aff}}$ to predicted MTL gains $\hat{y}_G^{\text{aff}}$ for a task group $G$:

$$\hat{y}_G^{\text{aff}} = f_{\text{non-linear}}\left(\phi\left(z_G\right), \omega_{\text{aff}}\right), \tag{9}$$

Both the hyperparameters of the B-spline expansion $\phi$ (i.e., spline degree and knot parameters) and the regression parameters $\omega_{\text{aff}}$ are optimized using cross-validation on a small training set of task groups $\mathcal{G}_{\text{train}}$, ensuring that the model generalizes well for inference on unseen task groups. We also explore alternative approaches for this non-linear transformation, which we discuss in Section E.

This first stage of our ensemble prediction framework bridges the gap between affinity scores and MTL gain values, which are correlated but may be on significantly different scales, and due to the non-linearity of the learned transformation, captures some of the higher-order task interactions.

### 3.2.2 DATA-DRIVEN RESIDUAL PREDICTION FOR IMPROVED ACCURACY

The second stage of our prediction framework improves upon the accuracy of the initial, first-stage predictions $\hat{y}_G^{\text{aff}}$ by learning to predict and reduce residual errors. The motivation for this second

stage is the need to capture higher-order interactions that pertain to particular tasks, which our first-stage predictor $f_{\text{non-linear}}$ cannot model. Therefore, we use our training set $\mathcal{G}_{\text{train}}$ to learn to predict residual errors based on which particular tasks are included in a group. Compared to purely black-box approaches like MTGNet, our residual predictor requires significantly less data since it learns to refine strong initial predictions instead of learning to predict MTL gains from scratch.

The residual prediction error for a group $G$ is $e_G^{\text{aff}} = y_G - \hat{y}_G^{\text{aff}}$, where $y_G$ denotes the ground-truth MTL gains, i.e., the residual error is the discrepancy between the ground-truth MTL gains and the first-stage predictions. Our goal is to learn a model that predicts $e_G^{\text{aff}}$ based on which particular tasks are included in group $G$. To this end, we train a supervised model $f_{\text{residual}}$ on training set $\mathcal{G}_{\text{train}}$ with labels $e_G^{\text{aff}}$, representing a group $G \subseteq \mathcal{T}$ as a *multi-hot encoded* vector $u_G$. That is, we represent group $G$ as a zero-one vector $u_G \in \{0, 1\}^{|\mathcal{T}|}$, where the element of $u_G$ corresponding to task $t \in \mathcal{T}$ is 1 if $t \in G$, and 0 if $t \notin G$. Our model $f_{\text{residual}}$ maps $u_G$ to a predicted residual, minimizing the loss between predicted and actual residuals for training groups $\mathcal{G}_{\text{train}}$:

$$\omega_{\text{res}} = \underset{\omega_{\text{res}}}{\arg\min} \sum_{G \in \mathcal{G}_{\text{train}}} \left( f_{\text{residual}}(u_G; \omega_{\text{res}}) - e_G^{\text{aff}} \right)^2 \tag{10}$$

where $\omega_{\text{res}}$ denotes the parameters of our model. We implement $f_{\text{residual}}$ as a ridge regression model:

$$\omega_{\text{res}} = \underset{\omega_{\text{res}}}{\arg\min} \frac{1}{2} \sum_{G \in \mathcal{G}_{\text{train}}} \left( e_G^{\text{aff}} - u_G^\top \omega_{\text{res}} \right)^2 + \lambda \|\omega_{\text{res}}\|^2. \tag{11}$$

$\lambda$ is a regularization hyperparameter that prevents overfitting by controlling the magnitude of the model coefficients $\omega_{\text{res}}$, which is particularly important when dealing with high-dimensional input data such as multi-hot encoding of task groups. By penalizing large coefficients, we ensure robust predictions. We can tune the value of hyperparameter $\lambda$ using cross-validation on training set $\mathcal{G}_{\text{train}}$.

We obtain the final MTL gain predictions for any group $G$ by combining the initial, first-stage predictions $\hat{y}_G^{\text{aff}}$ with the residual corrections predicted by $f_{\text{residual}}$:

$$\hat{y}_G^{\text{final}} = \hat{y}_G^{\text{aff}} + f_{\text{residual}}(u_G; \omega_{\text{res}}). \tag{12}$$

This refinement step enables us to capture the impact of particular tasks on MTL gains in a group, improving upon the first-stage predictions $\hat{y}_G^{\text{aff}}$. It is important to note that learning to predict these residual errors is easier than learning to predict MTL gains from scratch.

### 3.2.3 Group Selection based on MTL Gain Predictions

The selection of optimal task groups is a constrained search problem, which involves choosing a set of task groups that cover all $n$ tasks to maximize overall MTL gain (measured by, e.g., average MTL gain over all tasks) while limiting the number of selected groups to a budget $B$. While exhaustive or recursive search over the $2^n - 1$ combinations is feasible for small $n$, it quickly becomes infeasible as $n$ increases. Prior work has shown that despite this problem being generally NP-hard (through reduction from Set-Cover), it can be tackled using branch-and-bound strategies or binary integer programming solvers (Standley et al., 2020; Fifty et al., 2021; Zamir et al., 2018). So, following prior work, we also employ a branch-and-bound algorithm to optimize the selection of $B$ task groups, maximizing that the total predicted MTL gain across all tasks during inference.

## 4 Experiments and Results

To evaluate ETAP experimentally, we apply it to four diverse multi-task datasets (e.g., computer vision and time series) and compare it to various baseline approaches for predicting MTL gains. Our software implementation and the novel ridership dataset are available as supplementary material under open-source licenses.

### 4.1 Experimental Setup

**Dataset and MTL Architecture**   We evaluate ETAP on three diverse, widely used multi-task learning datasets, spanning vision (**CelebA**), time series (**ETTm1**), and molecular classification

Table 1: Multi-Task Learning Datasets and MTL Model Architectures

| Dataset | Domain, #Tasks ($|\mathcal{T}|$) | MTL Architecture | Notes |
|---|---|---|---|
| **CelebA** (Liu et al., 2015b) | Facial attributes, 9 | ResNet-18 | $\mathcal{T}$ from TAG Fifty et al. (2021) |
| **ETTm1** (Wu et al., 2021) | Temperature time-series, 7 | Autoformer | $\mathcal{T}$ from MTGNet Song et al. (2022) |
| **Chemical** (Jacob et al., 2008) | Molecule classification, 10 | FFNN encoder | $\mathcal{T} \subset 35$ with diverse sizes |
| **Ridership** (Zulqarnain et al., 2023) | Transit demand, 10 | LSTM + FFNN | Real-world ridership data |

(**Chemical**), and on a novel, real-world transit **Ridership** dataset—each with distinct task structures and MTL architectures that we selected to align with prior task-grouping studies. Table 1 summarizes the datasets and MTL model architectures; we provide additional details in Section B.1.

**Hyperparameter Tuning and Evaluation Setting** All ETAP hyperparameters are tuned via cross-validation on a small set of training groups to balance accuracy and efficiency. The initial prediction stage uses B-spline expansion of affinity scores followed by regularized linear regression, with spline degree, knot count, and regularization strength optimized (details are in Section B.2). Specifically, we tune the spline degree ($\in [2, 6]$), the number of knots (up to $\sqrt{|\mathcal{G}_{\text{train}}|}$), and the ridge-regularization strength ($\alpha \in [0.001, 1.0]$). The residual correction stage applies ridge regression with cross-validated regularization. Alternative models (e.g., KNN, Random Forest) are discussed in Section E. We compare ETAP with baselines in terms of MTL gain prediction and task group selection. All gain predictors are trained using randomized task subsets with different task groups, and evaluation is performed on a fixed hold-out test set, shared across all methods to ensure consistency (130 groups for CelebA, 40 for ETTm1, 230 for Chemical, and 200 for Ridership). For group selection evaluation, all possible combinations are considered for CelebA and ETTm1, while 50% of all combinations are considered for Chemical and Ridership datasets due to scalability constraints. To ensure robustness and reduce randomness, all training and evaluation are repeated (10× for gain prediction, 6× for group selection) with different random seeds. We report the average outcomes along with their respective confidence intervals. We evaluate prediction performance across all methods using the coefficient of determination ($R^2$) and Pearson correlation coefficients. For group selection, we assess total task performance across each datasets—classification error (CelebA), cross-entropy (Chemical), and mean-squared error (others).

## 4.2 EVALUATION OF THE PROPOSED FRAMEWORK

We evaluate ETAP with respect to three criteria: (1) how well our proposed task-affinity scores capture inter-task relationships; (2) our MTL gain predictions vs. established baselines; and (3) our selection of task groups vs. baseline approaches. We choose several representative baselines across affinity scoring, white-box and data-driven gain prediction, and MTL performance optimization. (i) **TAG** (Fifty et al., 2021) estimates MTL gains for higher-order task groups by averaging pairwise affinity scores obtained through a single MTL model with $n$ extra forward/backward passes per update; (ii) **GRAD-TAE** (Li et al., 2024) derives pairwise task affinities by estimating the performance of random task groups and averaging their resulting performance; (iii) **MTGNet** (Song et al., 2022) uses a self-attention transformer to predict MTL gains; (iv) **Linear Surrogate** (Li et al., 2023) predicts binary transfer outcomes in MTL. We also include (v) **PCGrad** (Yu et al., 2020), a gradient projection method orthogonal to our goal of pre-training group selection. Additionally, we include an **ablation study** in Section C, evaluating the contribution of each component of ETAP.

Table 2: Correlation between pairwise affinity scores and ground-truth MTL gains (higher values are better).

| Affinity Scoring Method | CelebA | ETTm1 | Chemical | Ridership |
|---|---|---|---|---|
| **TAG** (Fifty et al., 2021) | $0.16 \pm 0.00$ | $0.43 \pm 0.02$ | $0.33 \pm 0.19$ | $0.10 \pm 0.07$ |
| **GRAD-TAE** (Li et al., 2024) | $0.09 \pm 0.06$ | $-0.27 \pm 0.02$ | $0.28 \pm 0.06$ | $-0.22 \pm 0.19$ |
| **ETAP** | $0.32 \pm 0.00$ | $0.47 \pm 0.00$ | $0.40 \pm 0.03$ | $0.36 \pm 0.03$ |

**Prediction of Pairwise MTL Affinity** A key contribution of our approach is the proposed task-affinity score (Section 3.1.3). We compare our pairwise affinity scoring method to two state-of-the-art methods, TAG and GRAD-TAE. Table 2 compares their correlation with ground-truth pairwise MTL gains—higher correlation indicates better reflection of task dependencies, which is crucial for predicting group-level MTL gains. Since the affinity scores output by TAG are on a different

Table 3: Correlation between ground-truth and predicted MTL gains for groups (higher values are better).

| Method | Computational Cost | Correlation with Ground Truth | | | |
|---|---|---|---|---|---|
| | | CelebA | ETTm1 | Chemical | Ridership |
| **TAG** (Fifty et al., 2021) | * | $0.10 \pm 0.0$ | $0.47 \pm 0.0$ | $0.05 \pm 0.1$ | $0.15 \pm 0.1$ |
| **MTGNet** (Song et al., 2022) | $|\mathcal{G}_{\text{train}}| = 5$ | $0.10 \pm 0.2$ | $0.43 \pm 0.1$ | $0.22 \pm 0.1$ | $0.43 \pm 0.1$ |
| | $|\mathcal{G}_{\text{train}}| = 10$ | $0.22 \pm 0.1$ | $0.54 \pm 0.2$ | $0.34 \pm 0.2$ | $0.61 \pm 0.0$ |
| **ETAP** | $|\mathcal{G}_{\text{train}}| = 5$ | $0.41 \pm 0.2$ | $0.77 \pm 0.1$ | $0.40 \pm 0.1$ | $0.68 \pm 0.1$ |
| | $|\mathcal{G}_{\text{train}}| = 10$ | $0.45 \pm 0.1$ | $0.84 \pm 0.0$ | $0.50 \pm 0.1$ | $0.74 \pm 0.0$ |

scale than ground-truth MTL gains, their $R^2$ may be very low. To ensure that our comparison is fair, we compare the methods based on their correlation, which is independent of scale. While the MTL-performance estimation method of the GRAD-TAE approach can estimate MTL model performance reasonably well for groups, their pairwise task-affinity scoring approach struggles to align with ground-truth pairwise MTL gains. TAG achieves higher correlation, but incurs significant computational overhead. In contrast, ETAP achieves comparable correlation with actual pairwise affinities while substantially reducing runtime (46% on CelebA, 71% on ETTm1, 9% on Chemical, and 63% on Ridership; see Section D.1).

**Accuracy of MTL Gain Prediction for Groups**   Since ETAP integrates both white-box and data-driven components to predict MTL gains, we compare it with three baselines, **TAG**, **MTGNet**, and **Linear Surrogate**, in terms of prediction accuracy and data efficiency. TAG provides affinity scores to estimate MTL gains, but since these differ in scale from actual gains, a direct comparison based on $R^2$ would require rescaling. Instead, we report the correlation between predicted and ground-truth MTL gains, which is scale-invariant and ensures a fair evaluation. Table 3 shows correlations for **TAG**, **MTGNet**, and ETAP. TAG achieves moderate correlations but requires full MTL training with $n$ additional forward and backward passes for every task pair, making it computationally expensive. MTGNet is unstable with smaller training sets $\mathcal{G}_{\text{train}}$, performs better with larger ones, but fails to achieve consistently high correlations. In contrast, ETAP demonstrates strong and stable correlations even with limited training sets ($|\mathcal{G}_{\text{train}}| = 5$), achieving correlations of 0.41 on CelebA, 0.77 on ETTm1, 0.40 on Chemical, and 0.68 on Ridership. Correlation improves further with more training data. These findings underscore the efficiency and robustness of our approach, which outperforms both TAG and MTGNet while significantly reducing computational cost. ETAP also achieves higher correlation and F1 scores than **Linear Surrogate** (Section D.2) at the same computational cost on all datasets, e.g., improving F1 ($0.18 \rightarrow 0.31$) on ETTm1 and correlation ($0.49 \rightarrow 0.57$) on CelebA.

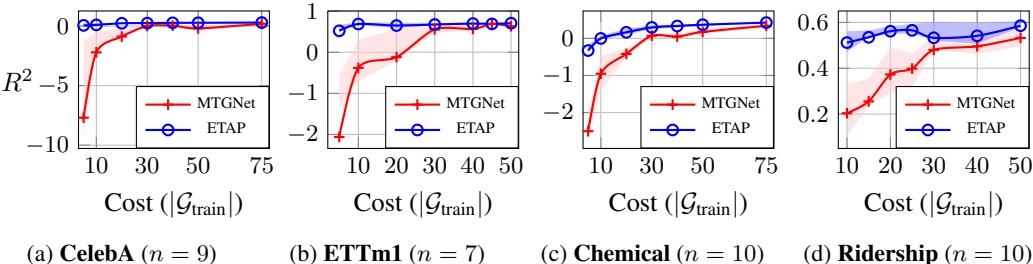

(a) **CelebA** ($n = 9$)    (b) **ETTm1** ($n = 7$)    (c) **Chemical** ($n = 10$)    (d) **Ridership** ($n = 10$)

Figure 2: Prediction performance ($R^2$) vs. computational cost ($|\mathcal{G}_{\text{train}}|$) for data-driven predictors, MTGNet and ETAP. MTGNet suffers from instability, with means outside the IQR due to outliers, whereas ETAP maintains consistency.

**Data Efficiency of MTL Gain Prediction**   We compare the computational cost of ETAP with the cost of **MTGNet**, a data-driven baseline, measuring computational cost as the number of MTL models trained to build training set $\mathcal{G}_{\text{train}}$. Figure 2 reports test-set $R^2$ values with interquartile range (IQR) for various training-set sizes $|\mathcal{G}_{\text{train}}|$. We finetuned the hyperparameters of MTGNet for each training set to optimize its performance. However, MTGNet exhibits highly random and unstable performance with small training sets, starting with large negative $R^2$ values and wide ranges. The mean $R^2$ value sometimes falls outside the upper or lower bounds due to extreme outliers, which highlights the need for a data-driven predictor that is reliable and stable. In contrast, ETAP consis-

tently yields positive $R^2$ values with narrow IQR across all datasets and training set sizes, indicating greater robustness.

Table 4: MTL performance (measured as total loss ($\pm$ std); lower is better) with tasks grouped based on various prediction methods. **Optimal** values are computed using an exhaustive search for CelebA and ETTm1, and using a search over a representative 50% of all task combinations for Chemical and Ridership.

| Dataset | #Groups | TAG | GRAD-TAE | MTGNet ($|\mathcal{G}_{train}|$) | ETAP ($|\mathcal{G}_{train}|$) | Optimal |
|---|---|---|---|---|---|---|
| **CelebA** ($|\mathcal{G}_{train}| = 10$) | 2 | $49.67 \pm 0.00$ | $50.78 \pm 0.59$ | $50.62 \pm 1.20$ | $49.92 \pm 0.23$ | $49.27$ |
| | 3 | $50.22 \pm 0.00$ | $50.78 \pm 1.93$ | $50.31 \pm 0.63$ | $49.61 \pm 0.30$ | $48.63$ |
| | 4 | $49.94 \pm 0.00$ | $52.83 \pm 1.70$ | $50.27 \pm 0.60$ | $49.41 \pm 0.25$ | $48.38$ |
| **ETTm1** ($|\mathcal{G}_{train}| = 10$) | 2 | $4.08 \pm 0.09$ | $4.08 \pm 0.06$ | $4.04 \pm 0.06$ | $4.02 \pm 0.05$ | $3.98$ |
| | 3 | $3.96 \pm 0.03$ | $4.04 \pm 0.06$ | $3.96 \pm 0.03$ | $3.93 \pm 0.07$ | $3.83$ |
| | 4 | $3.90 \pm 0.02$ | $4.01 \pm 0.04$ | $3.92 \pm 0.06$ | $3.89 \pm 0.09$ | $3.82$ |
| **Chemical** ($|\mathcal{G}_{train}| = 10$) | 2 | $4.69 \pm 0.11$ | $4.80 \pm 0.14$ | $4.79 \pm 0.31$ | $4.67 \pm 0.12$ | $4.56$ |
| | 3 | $4.80 \pm 0.06$ | $4.83 \pm 0.05$ | $4.89 \pm 0.15$ | $4.74 \pm 0.07$ | $4.52$ |
| | 4 | $4.95 \pm 0.08$ | $4.96 \pm 0.10$ | $4.94 \pm 0.09$ | $4.83 \pm 0.10$ | $4.67$ |
| **Ridership** ($|\mathcal{G}_{train}| = 10$) | 2 | $17.50 \pm 0.00$ | $18.48 \pm 0.80$ | $17.86 \pm 0.43$ | $17.77 \pm 0.26$ | $17.03$ |
| | 3 | $18.31 \pm 0.00$ | $17.94 \pm 0.47$ | $18.12 \pm 0.30$ | $17.83 \pm 0.26$ | $16.90$ |
| | 4 | $18.25 \pm 0.00$ | $18.45 \pm 0.36$ | $18.06 \pm 0.42$ | $17.59 \pm 0.19$ | $16.79$ |

**Prediction-based Group Selection** We evaluate ETAP against the baseline methods, **TAG**, **GRAD-TAE**, and **MTGNet**, in terms of the MTL performance of task groups selected based on their predictions. Using each method's MTL gain predictions, we apply a branch-and-bound algorithm to select task-groups from a candidate set of groups. For GRAD-TAE, we used the pairwise affinity values estimated by this method in the branch-and-bound algorithm. Table 4 reports MTL performances under various selection budgets, along with the MTL performance of optimal task grouping as reference points. The color gradients indicate relative performance, darker shades indicating better results. ETAP consistently achieves near-optimal performance, outperforming TAG and MTGNet, with lower variance. Even with a small training set, ETAP enables more effective grouping decisions and further improves with more data. Compared to **PCGrad** (Section D.3), ETAP consistently achieves lower loss—reducing error by up to 7.4% on ETTm1 (4.20 → 3.89) and 6.6% on Ridership (18.84 → 17.59)—highlighting the benefit of explicit grouping over implicit gradient-based conflict resolutions.

Table 5: Correlation and $R^2$ between ground-truth and predicted total MTL gains for groups (higher values are better).

| Dataset | $|\mathcal{G}_{train}|$ | Correlation with Ground Truth | | $R^2$ | |
|---|---|---|---|---|---|
| | | Ayman et al. (2023) | ETAP | Ayman et al. (2023) | ETAP |
| **Chemical** | 5 | $0.33 \pm 0.69$ | $0.74 \pm 0.04$ | $-0.84 \pm 2.12$ | $0.15 \pm 0.13$ |
| | 10 | $0.76 \pm 0.05$ | $0.75 \pm 0.05$ | $0.48 \pm 0.12$ | $0.36 \pm 0.05$ |
| | 20 | $0.78 \pm 0.06$ | $0.77 \pm 0.04$ | $0.56 \pm 0.12$ | $0.39 \pm 0.19$ |
| **Ridership** | 5 | $-0.01 \pm 0.13$ | $0.44 \pm 0.06$ | $-0.18 \pm 0.67$ | $0.16 \pm 0.08$ |
| | 10 | $0.05 \pm 0.14$ | $0.42 \pm 0.07$ | $-0.81 \pm 0.57$ | $0.12 \pm 0.09$ |
| | 20 | $0.13 \pm 0.08$ | $0.46 \pm 0.03$ | $-0.25 \pm 0.13$ | $0.18 \pm 0.05$ |

**Accuracy of Total MTL Gain Prediction for Groups** We also compare ETAP to the data-driven prediction method proposed by Ayman et al. (2023). Note that the method of Ayman et al. (2023) predicts the *total MTL gain for all tasks in a group*, instead of predicting *MTL gain for each task in a group* as ETAP and most other methods do (e.g., TAG and MTGNet). For a fair comparison, we adapted ETAP to the evaluation setting of Ayman et al. (2023): we used ETAP to predict MTL gain for each task in a group, and then we summed these gains up to obtain a predicted total MTL gain for the group. We present the results of these experiments in Table 5, which shows the correlation and $R^2$ between ground-truth and predicted total MTL gains (higher values are better). The results demonstrate that ETAP significantly outperforms this baseline in many cases.

Note that the computational cost of the method of Ayman et al. (2023) includes the substantial cost of training MTL models for all $\binom{n}{2}$ pairs of tasks. Further, the method also requires measuring ground-truth MTL gains for a set of training groups $\mathcal{G}_{train}$, which incurs the cost of training $|\mathcal{G}_{train}|$

MTL models for these groups. In contrast, ETAP achieves better accuracy at a lower cost: our method requires training only one MTL model to compute affinity scores (with an additional $\binom{n}{2}$ dot-product operations over already-computed gradient vectors, which incurs very low overhead) and measuring ground-truth MTL gains for a set of training groups $\mathcal{G}_{\text{train}}$. Since the number of training groups in most practical applications can be at least an order of magnitude lower than $\binom{n}{2}$, ETAP incurs a significantly lower overall computational cost.

## 5 RELATED WORK

A key challenge of multi-task learning is modeling task relationships and minimizing negative transfers between tasks. Early work (Argyriou et al., 2008; Caruana, 1997) showed that tasks with similarities or shared underlying structures benefit from joint training. More recent research has explored transferability from the perspectives of zero-shot learning (Pal & Balasubramanian, 2019), representation learning (Dwivedi & Roig, 2019), and information theory (Achille et al., 2021). Surveys by Zhang & Yang (2021) and Ruder (2017) review techniques for modeling task relatedness via shared features, low-rank structures, or clustering. As MTL applications scale, modeling task relatedness becomes critical not just for interpretability, but for guiding optimization either during training or through informed task grouping beforehand.

**MTL Optimization via Training Dynamics**  Gradient-based MTL strategies (e.g., PCGrad (Yu et al., 2020), GradNorm (Chen et al., 2018), Uncertainty Weighting (Kendall et al., 2018)) aim to mitigate task interference during training by optimizing shared training dynamics, rather than pre-selecting task groups. Our work is orthogonal to these MTL optimization methods since we focus on predicting group-level MTL gains in advance to guide task-grouping decisions prior to training.

**MTL Optimization via Pre-selecting Task Groups**  Recent research efforts have explored predicting MTL gains to optimize task grouping. Data-driven methods, such as HOA (Standley et al., 2020) and MTGNet (Song et al., 2022), require performing MTL training for many task groups to obtain ground-truth MTL gains, making them computationally expensive. Li et al. (2023) assume linear task interactions and introduce linear regression-based prediction models, which also require ground-truth gains for many task groups. Li et al. (2024) propose an alternative, GRAD-TAE: instead of directly measuring ground-truth gains, GRAD-TAE fine-tunes MTL hyperparameters on random task groups to estimate MTL performance. Although these estimates correlate well with the task performance of MTL, they remain resource intensive. Other approaches include analyzing task characteristics in NLP to understand MTL performance (Bingel & Søgaard, 2017) and meta-learning from task-specific features (Ayman et al., 2023); though these often underperform with limited data. Fifty et al. (2021) propose TAG, a white-box method that analyzes training dynamics through gradient-based affinity measurements, but requires multiple forward and backward passes per step, leading to high computational overhead. These methods, while offering low-variance estimates of pairwise affinities, may not fully capture complex task dependencies in larger groups.

## 6 CONCLUSION

Our proposed *Ensemble Task Affinity Predictor* (ETAP) framework addresses the limitations of existing approaches—instability in data-driven models and high computational cost in white-box methods—by refining low-cost affinity scores with data-driven prediction and residual correction, balancing computational efficiency and prediction accuracy. We demonstrated that by leveraging non-linear transformations and residual correction, ETAP achieves consistent, robust performance across multiple benchmark datasets, spanning diverse domains. Our framework lays a foundation for scalable MTL solutions that can optimize task grouping decisions across diverse domains, enabling more efficient multi-task learning applications.

**Acknowledgments**  This material is based upon work supported by the National Science Foundation (NSF) under Award No. CNS-1952011 and by the U.S. Department of Energy (DOE) under Award No. DE-EE0011188. Any opinions, findings and conclusions or recommendations expressed in this material are those of the author(s) and do not necessarily reflect the views of the NSF and the U.S. DOE. We would like to thank the anonymous reviewers of ICRL 2026 for their feedback on our work and their suggestions to improve our manuscript.

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

## A    NOTATION

To help readers navigate the theoretical framework and empirical findings effectively, Table 6 presents a comprehensive guide to the notation used throughout the paper.

Table 6: List of Notations

| Symbol | Description |
|---|---|
| $\mathcal{T}$ | Set of all tasks |
| $n$ | Number of tasks ($n = |\mathcal{T}|$) |
| $L_t$ | Loss of task $t \in \mathcal{T}$ |
| $L_{\mathcal{T}}$ | Total loss across all tasks ($L_{\mathcal{T}} = \sum_{t \in \mathcal{T}} L_t$) |
| $G$ | A task-group (i.e., a set of tasks $G \subseteq \mathcal{T}$ ) |
| $\mathcal{G}$ | A collection of task groups ($\mathcal{G} = \{G_1, G_2, \cdots\}$) |
| $\mathcal{X}$ | Input to any model (i.e., training/ test data for any task) |
| **Notations for STL and MTL Models** | |
| $\theta_{\text{STL}}$ | Parameters for an STL model trained on a single task $t \in \mathcal{T}$ |
| $\theta_{\text{MTL}}$ | Parameters for an MTL model trained on a group of tasks $G \subseteq \mathcal{T}$ |
| $y_{G \to t}$ | MTL gain for task $t$ in group $G$ ($y_{G \to t} = [L_t(\mathcal{X}, \theta_{\text{STL}}) - L_t(\mathcal{X}, \theta_{\text{MTL}})]/L_t(\mathcal{X}, \theta_{\text{STL}})$) |
| $\boldsymbol{y}_G$ | Vector of MTL gains for all tasks in a group $G$ |
| **Notations for Affinity Score Calculation** | |
| $z_{t_i \to t_j}^k$ | Task-affinity score of task $t_i$ on $t_j$ at gradient step $k$ |
| $\overline{z}_{t_i \to t_j}$ | Aggregated task-affinity score for task $t_i$ on $t_j$ across entire training |
| $\overline{z}_{G \to t_i}$ | Task-affinity score of a task-group $G$ on task $t_i$ ($\overline{z}_{G \to t_i} = \frac{1}{|G|-1} \sum_{t_j \in G, j \neq i} \overline{z}_{t_j \to t_i}$) |
| $\boldsymbol{z}_G$ | Affinity scores for all tasks within a group $G$ ($\{\overline{z}_{G \to t} : t \in G\}$). |
| **Notations for MTL Gain Predictor** | |
| $f_{\text{non-linear}}$ | Initial predictor function in ETAP (non-linear mapping) |
| $\phi$ | Non-linear transformation function |
| $\omega_{\text{aff}}$ | Regression coefficients for $f_{\text{non-linear}}$ |
| $\hat{\boldsymbol{y}}_G^{\text{aff}}$ | Prediction from $f_{\text{non-linear}}$ for a task group $G$ |
| $\boldsymbol{e}_G^{\text{aff}}$ | Error between actual and predicted MTL gains for task group $G$ ($\boldsymbol{y}_G - \hat{\boldsymbol{y}}_G^{\text{aff}}$) |
| $f_{\text{residual}}$ | Residual predictor function in ETAP (residual model) |
| $\omega_{\text{res}}$ | Model coefficients for $f_{\text{residual}}$ |
| $\boldsymbol{u}_G$ | Multi-hot encoding of the membership of task group $G$ ($\boldsymbol{u}_G \in \{0,1\}^{|\mathcal{T}|}$) |
| $\hat{\boldsymbol{y}}_G^{\text{final}}$ | Final MTL gain predictions for tasks $t$ in group, $G$ ($\hat{\boldsymbol{y}}_G^{\text{final}} = \hat{\boldsymbol{y}}_G^{\text{aff}} + f_{\text{residual}}(\boldsymbol{u}_G; \omega_{\text{res}})$). |

## B    EXPERIMENTAL SETUP

### B.1    DATASETS AND MTL ARCHITECTURES

We evaluate our proposed approach on four multi-task learning benchmarks from diverse domains, which had been used to assess task grouping strategies in prior MTL studies.

**CelebA** Liu et al. (2015b) is a large-scale dataset of face images with multiple attributes, commonly used in MTL research Ehrlich et al. (2016); Guo et al. (2020); Han et al. (2017). The Task-Affinity Grouping (TAG) approach Fifty et al. (2021) evaluated task combinations on a subset of nine tasks from CelebA. We adopt the same dataset and use the optimized ResNet-18 architecture He et al. (2016) as the baseline MTL model, following the experimental setup in prior work Fifty et al. (2021).

**ETTm1** (Electricity Transformer Temperature) Wu et al. (2021) is an electric load dataset comprising seven time-series forecasting tasks which are used to evaluate MTGNet approach Song et al. (2022). We follow prior work Wu et al. (2021), framing it as a multivariate time-series forecasting problem, where each series forecast is treated as a separate task. The Autoformer Wu et al. (2021) model serves as the MTL architecture with an SGD optimizer.

**Chemical (MHC-I)** Jacob et al. (2008) is a dataset designed to classify peptide binding affinities for different molecules. The dataset contains 35 tasks, each representing a molecule's binding classification. We select 10 tasks with various numbers of training samples for our experiments. Each task has 180 binary input features. We implement feed-forward neural networks for both multi-task learning (MTL) and single-task learning (STL), using a shared encoder followed by task-specific

output layers. The shared encoder consists of two fully connected layers with 32 and 16 *ReLU* units, respectively, capturing a common representation across tasks. Each task-specific decoder contains a single *sigmoid*-activated output neuron. Models are trained using binary cross-entropy loss and optimized with stochastic gradient descent (learning rate = 0.001, momentum = 0.9). This single architecture is used consistently for all STL and MTL runs.

**Ridership**: Alongside these widely used benchmarks, we also employ a novel, real-world transit ridership dataset from a U.S. city, curated by Zulqarnain et al. (2023), containing stop-level passenger counts for buses, enriched with predictive features. Each task is to predict bus ridership for a specific route and direction. We select 10 such tasks with various data availability, from data-rich to highly sparse. We use a hybrid multi-input MTL architecture to model ridership prediction across routes. The predictive features are of three types: (i) sequential statistics from the past 10 trips (e.g., median/max load, time gaps), (ii) numerical features such as weather, delays, and service headway, and (iii) categorical features like time-of-day and calendar indicators, embedded via learnable layers (embedding dim = 4). The shared encoder combines stacked Bidirectional LSTMs (64 units each) for temporal features, feed-forward layers for numerical features, and embeddings for categorical inputs, projecting to a 16-dimensional latent space. Each task-specific decoder contains two dense layers (8 and 1 units) for final regression. A shared encoder captures common structure across tasks, while lightweight task-specific decoders generate the final predictions. Models are trained with Adam (learning rate = 0.001), using MSE loss, dropout ($0.2 - 0.3$), and batch normalization for regularization.

Please note that the MTL architectures are not a critical component of our approach and serve only to enable experimentation with predicting MTL gains and grouping tasks.

### B.2 Hyperparameter Search

All ETAP hyperparameters are tuned via cross-validation on randomly selected subsets of training groups. The first stage of ETAP employs a regularized linear regression model over a B-spline transformation of affinity scores. We perform a grid search for spline degree (2–6), number of knots (up to $\sqrt{|\mathcal{G}_{\text{train}}|}$ samples, where $\mathcal{G}_{\text{train}}$ indicates the number of groups for training the predictor), and Ridge regularization strength ($\alpha$ ranging from $0.001$ to $1.0$). The residual correction stage uses Ridge regression with $\alpha$ selected via cross-validation over a similar range. For each training instance of the gain predictor, the best hyperparameters are determined using cross-validation over training groups only, and then fixed for prediction on unseen groups. To ensure robustness and minimize the impact of randomness, all training and evaluation procedures are repeated with different random seeds: 10× for gain prediction and 6× for group selection. The seeds are randomly chosen, and the specific seeds used to generate the results reported in the main text are documented in the code appendix.

For the **MTGNet** baseline Song et al. (2022), we conduct a separate hyperparameter search over the number of Transformer layers and the embedding dimension, selecting from values in the range of 4 to 64. For further architectural details, we refer readers to the code implementation provided in Song et al. (2022). All models are trained and validated using only training groups, ensuring fair generalization to held-out test groups. To replicate **TAG** Fifty et al. (2021) and **PCGrad** Yu et al. (2020), we use the official software implementations provided by the original authors.

**Hardware Configuration** All model training (including both single-task and multi-task experiments) is performed on a Unix-based high-performance server running Ubuntu 22.04.5 LTS. The system is equipped with two NVIDIA RTX A5000 GPUs (24 GB each, CUDA 12.8). GPU acceleration is utilized for training where applicable.

## C Ablation

To evaluate the contribution of each component in ETAP, we perform an ablation study, focusing particularly on the residual prediction step. By removing this step and comparing the resulting performance to the entire method, we can evaluate its role in improving MTL gain predictions and refining task-group estimates.

## C.1 Contribution of Residual Prediction

To investigate the significance of incorporating a residual prediction step, we train a model without this step and evaluate it under identical conditions as the full framework. The comparison highlights how this step influences prediction accuracy and its alignment with ground-truth MTL gains.

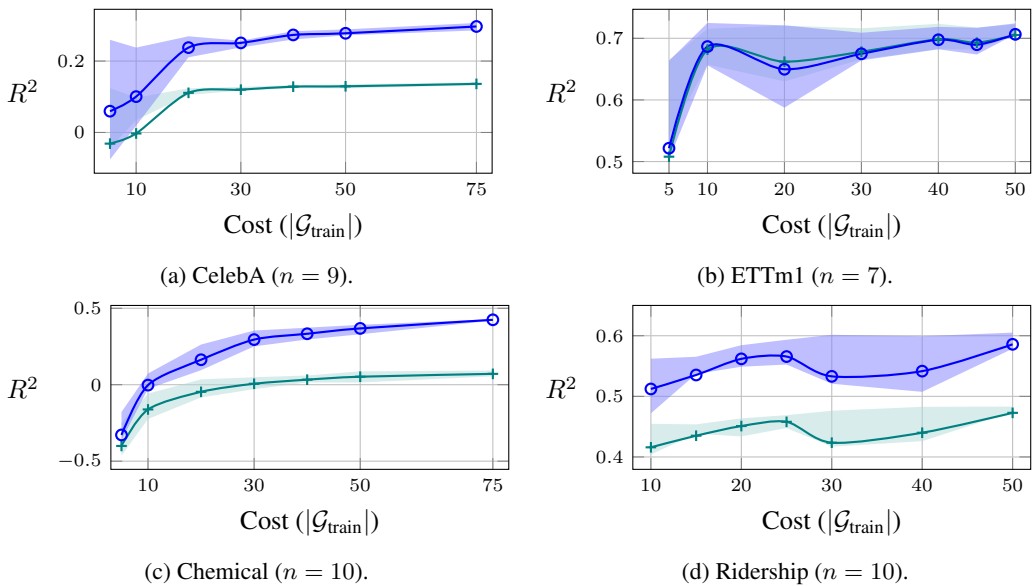

(a) CelebA ($n = 9$).

(b) ETTm1 ($n = 7$).

(c) Chemical ($n = 10$).

(d) Ridership ($n = 10$).

Figure 3: Comparison of prediction performance ($R^2$) between the initial predictions (without residual adjustment ✛) and the final predictions (with residual adjustment ⊕) across four datasets. The residual adjustment step improves accuracy, particularly on CelebA, Chemical, and Ridership, with a slight improvement observed on ETTm1.

Figure 3 shows a comparison in terms of prediction performance ($R^2$-values) on the test set groups. The results show that residual prediction significantly improves accuracy for the CelebA, Chemical, and Ridership datasets, with higher alignment between predicted and ground-truth MTL gains.

Table 7: Correlation with ground-truth MTL gains across different computational costs ($|\mathcal{G}_{\text{train}}|$) for the predictor on all benchmarks (higher values are better).

| Dataset | $|\mathcal{G}_{\text{train}}|$ | Correlation (Predicted vs. Actual) | |
|---|---|---|---|
| | | Initial | Final |
| CelebA | 5 | $0.27 \pm 0.15$ | $0.41 \pm 0.16$ |
| | 10 | $0.31 \pm 0.13$ | $0.45 \pm 0.13$ |
| ETTm1 | 5 | $0.75 \pm 0.15$ | $0.77 \pm 0.12$ |
| | 10 | $0.84 \pm 0.02$ | $0.85 \pm 0.02$ |
| Chemical | 5 | $0.32 \pm 0.06$ | $0.40 \pm 0.07$ |
| | 10 | $0.36 \pm 0.03$ | $0.50 \pm 0.07$ |
| Ridership | 10 | $0.66 \pm 0.02$ | $0.71 \pm 0.03$ |
| | 15 | $0.67 \pm 0.03$ | $0.73 \pm 0.04$ |

When comparing initial predictions based on task affinity scores and non-linear mapping with those enhanced through residual correction, the differences in some cases are marginal. However, this is not a limitation of the approach. Even in cases where both approaches achieve similar performance, our suggested method generally maintains a stronger overall correlation with the ground-truth MTL gains and outperforms baseline approaches across all benchmarks as shown in Table 7. This consistency demonstrates the framework's robustness, particularly its ability to handle complex and non-linear task dependencies at a much lower cost. The residual correction step provides a crucial mechanism to address systematic errors in initial gain estimates, leading to more reliable predictions for MTL grouping.

## C.2 CONTRIBUTION OF NON-LINEAR MAPPING ON AFFINITY SCORES

To assess the impact of the non-linear transformation in ETAP, we compare ETAP's performance against a variant that uses an affine mapping to align task affinity scores with ground-truth MTL gains. We aim to determine whether the added expressiveness of non-linear mappings (e.g., B-spline) meaningfully improves prediction accuracy over an affine mapping.

Table 8: Effect of non-linear transformation on task-affinity scores: comparison of affine vs. B-spline mapping in terms of $R^2$ and correlation with ground-truth MTL gains.

| Dataset | $R^2$ (Affine) | $R^2$ (B-spline) | Corr. (Affine) | Corr. (B-spline) |
|---|---|---|---|---|
| CelebA | $0.13 \pm 0.00$ | $0.13 \pm 0.01$ | $0.37 \pm 0.00$ | $0.37 \pm 0.01$ |
| ETTm1 | $0.18 \pm 0.02$ | $0.68 \pm 0.05$ | $0.45 \pm 0.00$ | $0.84 \pm 0.02$ |
| Chemical | $-0.06 \pm 0.11$ | $-0.04 \pm 0.11$ | $0.35 \pm 0.00$ | $0.37 \pm 0.04$ |
| Ridership | $0.35 \pm 0.03$ | $0.42 \pm 0.04$ | $0.62 \pm 0.00$ | $0.67 \pm 0.02$ |

As shown in Table 8, the non-linear B-spline transformation consistently matches or outperforms the affine baseline in terms of both $R^2$ and correlation across all four datasets. The gains are especially pronounced on ETTm1 and Ridership, where complex, non-linear relationships exist between affinity and MTL gain. These results confirm that MTL gains are not linearly explained by affinity scores alone, and that the non-linear mapping plays a key role in modeling this complexity. Overall, the B-spline transformation enhances ETAP's ability to predict MTL gains by capturing non-linear patterns that an affine transformation cannot express, making it an essential component of the pipeline.

# D ADDITIONAL RESULTS

## D.1 RUN-TIME COMPARISON FOR PREDICTION OF PAIRWISE MTL AFFINITY

To predict pairwise multi-task learning affinity, we compute task-affinity scores that estimate the compatibility between task pairs. In this section, we compare the run-time performance of our affinity scoring approach (introduced in Section 3.1) with the Inter-Task Affinity (ITA) score used in Task-Affinity Grouping (TAG) Fifty et al. (2021).

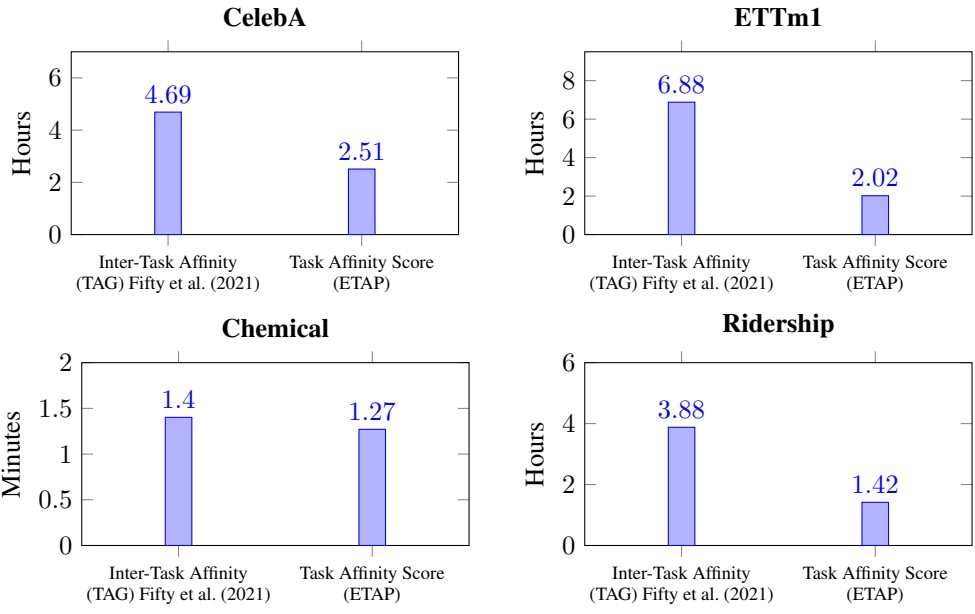

Figure 4: Runtime for CelebA and ETTm1 datasets (measured in hours) and the Chemical dataset (measured in minutes). Running times include task-affinity scores calculations and complete multi-task learning training for all tasks together in the benchmark using baseline MTL models.

Figure 4 presents a side-by-side comparison of the computational efficiency of the two methods, evaluated under identical hardware and software conditions for fairness. Our approach consistently achieves lower run-times across all datasets—CelebA, ETTm1, Chemical, and Ridership. For instance, on CelebA, TAG requires 6.88 hours to compute affinity scores, whereas our method completes the process in just 2.51 hours. Similar improvements are observed on other datasets: a 71% reduction in runtime on ETTm1, 9% on Chemical, and 63% on Ridership.

These results highlight the scalability of our approach and its ability to deliver robust affinity predictions with significantly reduced computational overhead.

## D.2 COMPARISON WITH LINEAR SURROGATE MODEL FOR GAIN PREDICTION

We also compare our proposed method, ETAP, with the Linear Surrogate model proposed by Li et al. (2023) in terms of prediction accuracy. Linear surrogate approach was originally designed for predicting binary transfer outcomes (positive vs. negative transfer) in MTL. To ensure a fair comparison, we train both ETAP and the surrogate baseline on the same set of task groups across four benchmarks.

Since the approach of Li et al. (2023) focuses on binary classification of transfer gains, we evaluate both methods using two metrics:

- **Correlation:** Pearson correlation between predicted and ground-truth MTL gains.
- **F1 Score:** Computed on the binary classification task of predicting the correct sign of the MTL gain.

Table 9: ETAP vs. Linear Surrogate Model Li et al. (2023) for identifying positive vs. negative transfer.

| Dataset | Training Cost ($|\mathcal{G}_{\text{train}}|$) | Surrogate Model | | ETAP | |
|---|---|---|---|---|---|
| | | Corr. | F1 | Corr. | F1 |
| **CelebA** | 32 | 0.49 | 0.61 | **0.57** | **0.70** |
| **ETTm1** | 21 | 0.35 | 0.18 | **0.64** | **0.31** |
| **Chemical** | 37 | 0.57 | 0.92 | **0.62** | **0.94** |
| **Ridership** | 35 | 0.67 | 0.81 | **0.77** | **0.85** |

As shown in Table 9, ETAP consistently outperforms the surrogate modeling approach Li et al. (2023) across all datasets in both correlation with ground-truth MTL gains and F1 score for classifying positive vs. negative transfer. The performance gap is particularly notable on ETTm1, where ETAP achieves a correlation of 0.64 and an F1 score of 0.31, significantly surpassing the surrogate model's weaker correlation (0.35) and low F1 score (0.18). On CelebA and Ridership, ETAP also improves both metrics, demonstrating better alignment with ground-truth and stronger classification of transfer outcomes. Even on Chemical, where the surrogate model performs relatively well, ETAP offers a modest but consistent improvement. These results highlight ETAP's superior ability to capture complex, non-linear, and higher-order task interactions—capabilities that the surrogate model, constrained by its linear assumption, fails to match.

## D.3 COMPARISON WITH PCGRAD FOR OPTIMIZED MTL PERFORMANCE

To further assess the effectiveness of ETAP, we include a comparison with PCGrad Yu et al. (2020), a gradient-projection-based method designed to resolve conflicts between task gradients during training. Unlike our grouping-based strategy, PCGrad assumes a single unified task set (i.e., one group that includes all tasks) and operates entirely in the gradient space, making it orthogonal to our approach.

Table 10 presents MTL performance (lower is better) across four diverse datasets. ETAP outperforms both Naive MTL (which jointly trains all tasks without grouping) and PCGrad in most cases, especially when tasks are split into 3–4 groups. For example, on the ETTm1 dataset, ETAP with 4 groups achieves a loss of 3.89, significantly lower than 4.20 (Naive MTL) and 4.19 (PCGrad). Similar trends are observed for Ridership, where ETAP reduces loss to 17.59, compared to 18.95 and 18.84.

Table 10: Comparison of multi-task learning performance (measured in loss; lower is better) across different methods (Naive MTL, PCGrad, and our proposed ETAP method with 2, 3, and 4 task-group splits). Results are reported as mean $\pm$ standard deviation across multiple runs.

| Dataset | Naive MTL | PCGrad | ETAP (SPLITS) | | |
|---|---|---|---|---|---|
| | | | 2 | 3 | 4 |
| CelebA | $50.02 \pm 0.4$ | $49.99 \pm 0.5$ | $49.92 \pm 0.2$ | $49.61 \pm 0.3$ | $49.41 \pm 0.3$ |
| ETTm1 | $4.20 \pm 0.0$ | $4.19 \pm 0.0$ | $4.02 \pm 0.1$ | $3.93 \pm 0.1$ | $3.89 \pm 0.0$ |
| Chemical | $4.83 \pm 0.2$ | $6.20 \pm 0.2$ | $4.67 \pm 0.0$ | $4.74 \pm 0.2$ | $4.83 \pm 0.2$ |
| Ridership | $18.95 \pm 0.7$ | $18.84 \pm 0.3$ | $17.77 \pm 0.3$ | $17.83 \pm 0.3$ | $17.59 \pm 0.2$ |

Interestingly, on the Chemical dataset, PCGrad underperforms Naive MTL, suggesting that gradient projection alone may not resolve deeper incompatibilities between certain tasks. Meanwhile, ETAP maintains competitive performance by grouping synergistic tasks. On CelebA, all methods yield comparable results, though ETAP still benefits from modest improvements with more refined groupings.

These results reinforce that task grouping can capture task relationships beyond gradient conflict and may complement or even outperform purely optimization-based strategies like PCGrad.

## D.4 STATISTICAL SIGNIFICANCE

To assess whether the performance differences between ETAP and the baselines are statistically significant, we conduct a paired Wilcoxon signed-rank test across all datasets and training sample

Table 11: Wilcoxon signed-rank test results comparing MTGNet vs. ETAP across datasets and training sample sizes. Bold $p$-values indicate statistical significance at $p < 0.05$.

| Dataset | Computational Cost | Sample Size | Wilcoxon $p$-value |
|---|---|---|---|
| CelebA | 5 | 10 | **0.0020** |
| | 10 | 10 | **0.0039** |
| | 20 | 10 | **0.0039** |
| | 30 | 10 | **0.0020** |
| | 40 | 10 | **0.0039** |
| | 50 | 10 | **0.0020** |
| | 75 | 10 | **0.0098** |
| ETTm1 | 5 | 10 | **0.0020** |
| | 10 | 10 | **0.0020** |
| | 20 | 10 | **0.0137** |
| | 30 | 10 | **0.0137** |
| | 40 | 10 | **0.0195** |
| | 45 | 10 | 0.8457 |
| | 50 | 10 | **0.0059** |
| Chemical | 5 | 10 | **0.0020** |
| | 10 | 10 | **0.0039** |
| | 20 | 10 | **0.0020** |
| | 30 | 10 | **0.0039** |
| | 40 | 10 | **0.0039** |
| | 50 | 10 | **0.0020** |
| | 75 | 10 | **0.0020** |
| Ridership | 10 | 10 | **0.0020** |
| | 15 | 10 | **0.0020** |
| | 20 | 10 | **0.0020** |
| | 25 | 10 | **0.0020** |
| | 30 | 10 | 0.1602 |
| | 40 | 10 | 0.2304 |
| | 50 | 10 | **0.0137** |

sizes. This non-parametric test evaluates whether the distribution of prediction performance (e.g., $R^2$ or correlation with ground-truth MTL gains) differs consistently between two approaches.

Table 11 shows whether the differences between ETAP and MTGNet are statistically significant in terms of prediction performance ($R^2$). For each configuration, we collect paired performance values from 10 randomized trials and compute a $p$-value using the Wilcoxon signed-rank test. $p$-value tests the null hypothesis that the two models perform equally, considering deviations in either direction (i.e., whether ETAP is better or worse). Results are reported in Table 11, where statistically significant differences ($p < 0.05$) are shown in bold. Across most settings, ETAP significantly outperforms MTGNet, particularly at *lower computational costs*. This confirms the robustness of the observed improvements and suggests that the gains from ETAP are unlikely to be due to random chance.

Table 12: Wilcoxon signed-rank test $p$-values comparing TAG vs. ETAP across benchmarks.

| Dataset | Sample Size | Cost (ETAP) | Wilcoxon $p$-values |
|---------|-------------|-------------|---------------------|
| CelebA | 7 | 5 | **0.0313** |
| ETTm1 | 7 | 5 | **0.0313** |
| Chemical | 7 | 5 | **0.0156** |
| Ridership | 7 | 10 | **0.0078** |

As shown in Table 12, we assess the statistical significance of the difference in gain prediction performance between TAG and ETAP. Since TAG is an unsupervised method that does not rely on training samples, we compare it against ETAP at a fixed training size ($|\mathcal{G}_{\text{train}}| = 5$ or $10$ depending on the benchmark), using correlation with ground-truth MTL gains across multiple random seeds. A Wilcoxon signed-rank test reveals that ETAP significantly outperforms TAG on all benchmarks ($p < 0.05$).

# E    ALTERNATIVE COMPONENTS FOR ETAP FRAMEWORK

For our ensemble framework, ETAP, we experiment with different components for both stages of the ensemble.

## E.1    APPROACHES FOR INITIAL PREDICTION

The main text presents results for B-spline transformation followed by a regularized linear regression. Here, we present alternative methods tested for the initial prediction process.

**K-nearest neighbor (KNN):** We use a KNN regressor to predict MTL gains by determining the optimal number of neighbors ($k$). The input features, derived from pairwise task affinities, are scaled before training. We use cross-validation to select $k$ by minimizing the mean squared error (MSE).

**Random Forest Regression (RF):** We utilize a Random Forest regressor Breiman (2001) with task-affinity-based predictions as input features. Hyperparameters such as the number of trees, tree depth, and minimum sample requirements are optimized via a grid search with cross-validation based on MSE.

For all approaches, we implement the models using the standard functions from `Sci-kit Learn` Pedregosa et al. (2011). We determine the optimal hyper-parameters for the models using grid search and cross-validation on the training groups, where the respective models are evaluated based on the mean squared error (MSE) as the evaluation metric. The best models are selected based on cross-validation performances. Once the optimal hyper-parameters are identified, the models are trained on the entire training data and used to make predictions on both the training and test sets, which are later utilized in the residual prediction step.

**Results**    Figure 5 illustrates the performance of various approaches applied to the first step of an ensemble model on all datasets. These approaches include our suggested method, *BSplines Basis*

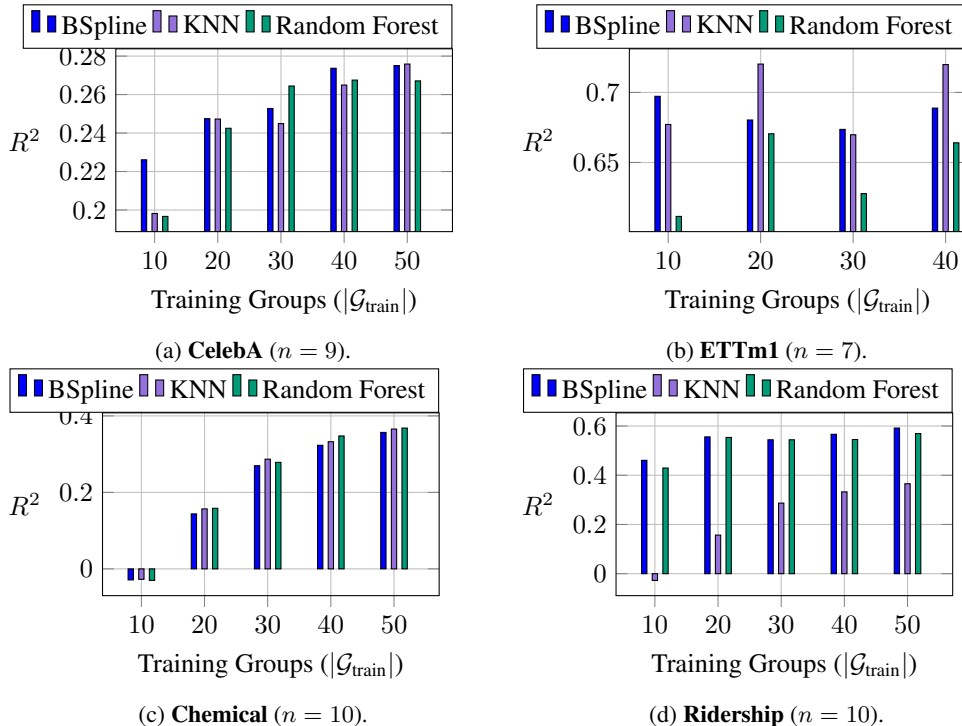

Figure 5: Comparison of prediction performance ($R^2$) for different approaches applied in the first step of the ensemble model ($f_{\text{non-linear}}$) on the CelebA, ETTm1, Chemical, and Ridership datasets. These performances are reported after applying a residual correction step ($f_{\text{residual}}$), where regularized ridge regression is used for residual prediction.

*Expansion followed by Regularized Linear regression*, chosen for its simplicity, efficiency, and interpretability. Alternate methods, such as the KNN, and Random Forest with Ridge regression as the residual predictors are also evaluated. Performance is assessed using $R^2$ scores across various training sample sizes, shedding light on how each method generalizes in different scenarios.

For CelebA and Ridership dataset, BSplines shows relatively stronger performance with all training sample sizes, indicating its ability to model non-linear task relationships effectively with sufficient training groups. For ETTm1, both B-splines with Ridge and KNN maintain robust and stable $R^2$ scores across sample sizes, with KNN slightly outperforming B-splines at larger training sizes, highlighting their capacity to balance complexity and regularization in smaller datasets. For the Chemical dataset, all approaches achieve comparable performance, with consistently higher $R^2$ scores at larger training sizes.

While Random Forest delivers strong performance, its lack of interpretability and poor handling of sparse data make it less suitable for our needs. Although KNN also performs competitively, especially in certain datasets, its high memory requirements and limited scalability with increasing data size make it a less practical choice in real-world applications. These factors collectively support our rationale for suggesting BSplines with regularized linear regression as the initial affinity predictor, as it balances model simplicity, computational efficiency, and generalization performance.

### E.2 APPROACHES FOR RESIDUAL PREDICTION

For the second stage of our framework, which focuses on residual error correction, we use ridge regression with the regularization parameter optimized via cross-validation. We also explore another boosted regression method.

**Extreme Gradient Boosting (XGBoost)** For residual correction, we employ an XGBoost regressor Chen & Guestrin (2016) to predict the residual errors between the initial task-affinity-based predictions and the actual MTL gains. The input features are zero-one vectors representing task

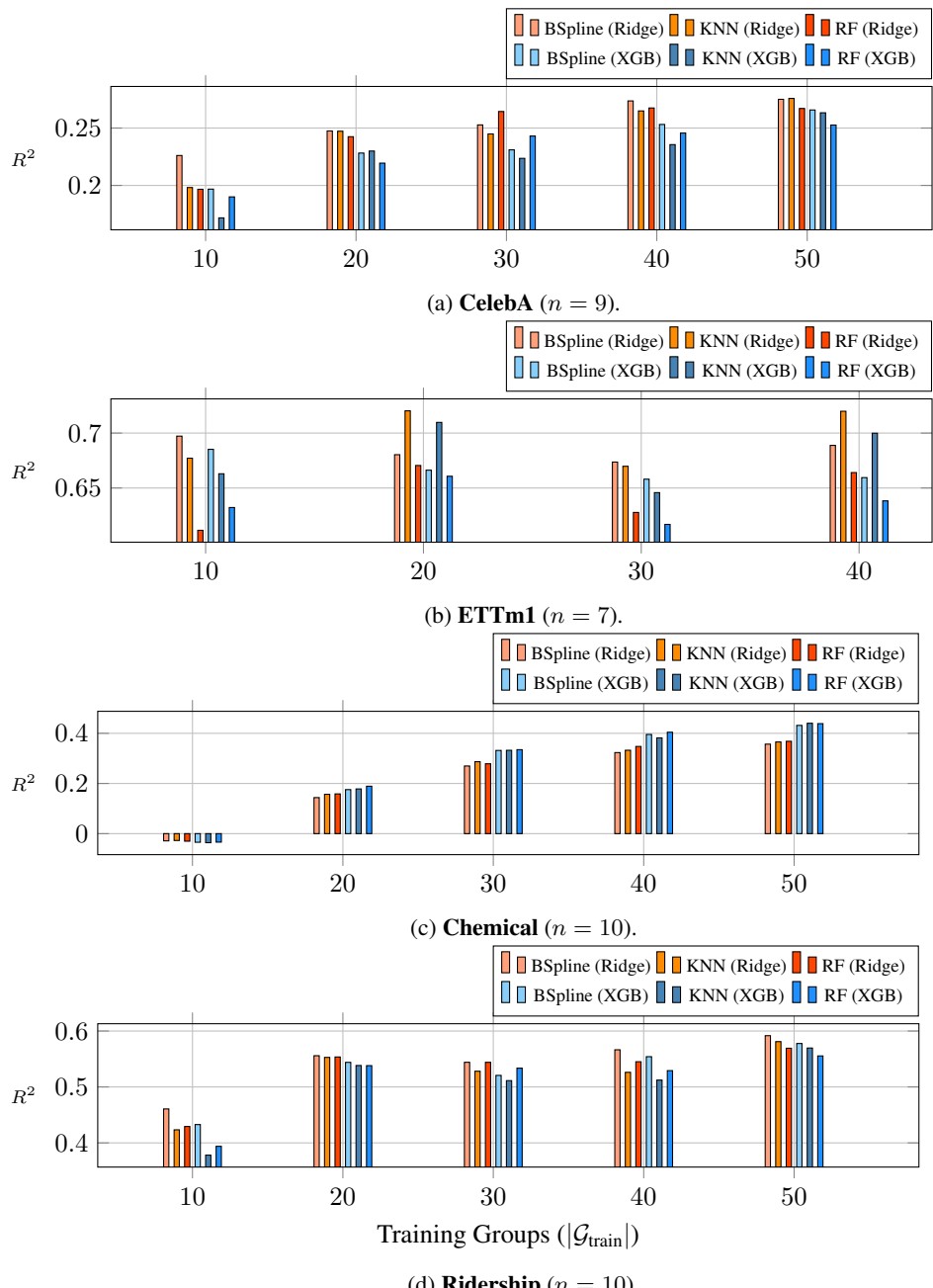

Figure 6: Prediction performance ($R^2$) for different approaches applied in the residual learning step of the ensemble model ($f_{\text{residual}}$) on the CelebA, ETTm1, Chemical, and Ridership datasets. Results are reported for both Ridge and XGB-based residual predictors applied to the initial prediction methods: BSpline, KNN, and Random Forest (RF).

groups, while the labels are the corresponding residual values. We perform hyperparameter tuning using a grid search with cross-validation, optimizing parameters such as the number of estimators, tree depth, learning rate, and subsampling ratios. The model is trained on scaled input data, and predictions are made on training and test sets. This method refines initial predictions by capturing complex patterns in residuals, thereby enhancing MTL gain prediction accuracy.

**Results**    Figure 6 shows the performance of residual models across all benchmarks, evaluated with varying numbers of training groups. After generating initial predictions using task-affinity scores

and non-linear mapping, we compare two residual predictors: Ridge regression and XGB, each represented by distinct color gradients in the figure.

Ridge regression consistently achieves high $R^2$ scores across different datasets and training sizes, demonstrating strong generalization and stability. In contrast, XGB exhibits greater variability, particularly with smaller training groups. For instance, while XGB slightly outperforms Ridge on the Chemical dataset with larger training sizes, the difference is negligible when fewer groups are available. Across CelebA, ETTm1, and Ridership, Ridge regression delivers better average performance, whereas XGB's sensitivity to dataset size and stability makes it less reliable under limited supervision.

Given its robustness, simplicity, and resistance to overfitting, we recommend *Ridge regression* for residual correction. Its regularization ensures stable improvements even with scarce training data, in contrast to XGB, which can be more sensitive to hyperparameters and noise.

# F   VARIABILITY OF PER-STEP AFFINITY SCORE DURING TRAINING

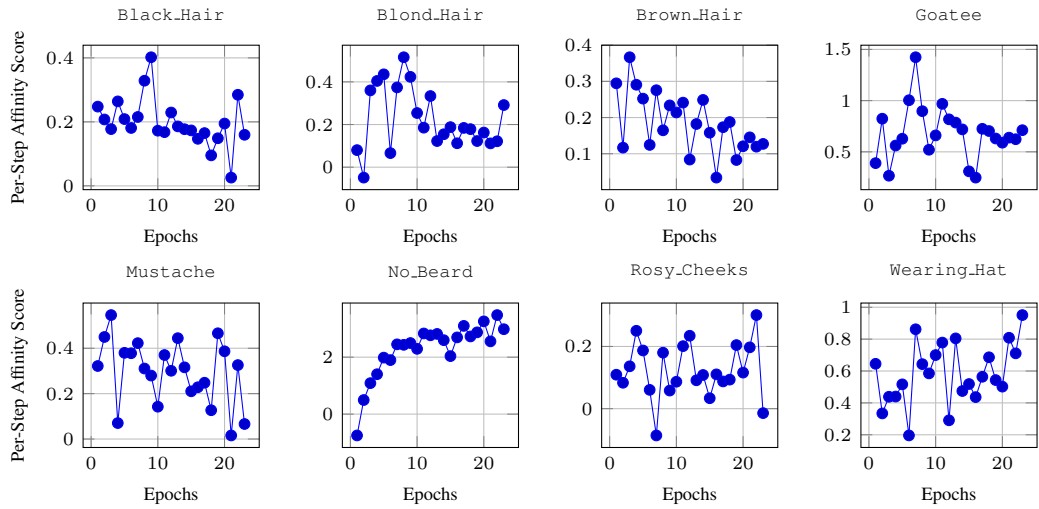

Figure 7:   Evolution of per-step affinity score $z^k_{t_i \to t_j}$ during training between a source task $i$ (5_o_Clock_Shadow) and each target task $j$ (Black_Hair, Blond_Hair, Brown_Hair, Goatee, Mustache, No_Beard, Rosy_Cheeks, Wearing_Hat) in dataset CelebA.

