# OpenReview forum: "Ensemble Prediction of Task Affinity for Efficient Multi-Task Learning"
_ICLR.cc/2026/Conference — ICLR 2026 Poster_

### Official Review · Reviewer_66hX · 2025-10-26

**Soundness:** 2
**Presentation:** 3
**Contribution:** 2
**Rating:** 6
**Confidence:** 4

**Summary:**

This paper proposes ETAP (Ensemble Task Affinity Predictor), a framework for predicting multi-task learning (MTL) gains to enable efficient task grouping. The approach combines white-box gradient-based affinity scoring with data-driven ensemble prediction.

**Strengths:**

1. The two-stage ensemble design that uses gradient-based affinity scores as foundation and refining with data-driven models is reasonable, it combines white-box gradient-based affinity scoring with data-driven ensemble prediction.
2. ETAP achieves impressive runtime reduction while maintaining or improving correlation with ground-truth gains. This is a meaningful practical contribution.

**Weaknesses:**

1. The gradient-based affinity score (Equation 5) is quite similar to existing work (TAG), just removing the auxiliary forward/backward passes. The B-spline transformation and ridge regression are standard techniques. The main novelty seems to be in combining these pieces, which feels somewhat incremental. Can you clarify what is fundamentally new here beyond engineering different existing methods together?
2. While you claim ETAP is "scalable," all experiments use relatively small task sets (n=7-10). What happens when n=50 or n=100? The affinity score computation still requires training one MTL model with all tasks, and the pairwise scores grow as O(n²). The paper doesn't really demonstrate scalability to large task pools that appear in real applications.
3. Section 3.2.3 uses branch-and-bound from prior work. So the contribution is really just the gain prediction, not the actual grouping algorithm. This should be made more clear in the contribution claims.

**Questions:**

See weaknesses.

---

> ### Author Response · Authors · 2025-11-21
>
> ## W1: Clarification of Novelty
> We agree that certain individual components of ETAP, such as gradient-based affinity and ridge regression, exist in isolation in prior work. However, the novelty of ETAP lies not only in the careful engineering of these components, but mainly in **bridging the gap between gradient-based and data-driven approaches for predicting MTL gains**. By introducing this novel paradigm, ETAP addresses the fundamental limitations of both gradient-based and data-driven approaches.
>
> On the one hand, prior gradient-based affinity methods (e.g., TAG) treat gradient alignment as the final predictor, which suffers from substantial issues: (i) a scale mismatch between affinity scores and MTL gains, (ii) high prediction bias due to an implicit linearity assumption, and (iii) an inability to capture higher-order, group-dependent interactions. On the other hand, prior data-driven methods (e.g., MTGNet and Linear Surrogate) treat MTL as a black box, relying only on measured MTL gains to make predictions, which also suffers from substantial issues: (i) high prediction variance due to a lack of informative priors and (ii) poor, brittle performance when the number of training groups is low.
>
> ETAP uniquely bridges these two perspectives by introducing a principled two-stage prediction ensemble that combines gradient-based affinity (i.e., white-box approach) with data-driven prediction (i.e., black-box approach). Adopting this paradigm requires a number of innovations: ETAP first computes **low-cost gradient-based affinity scores** using a novel method, which attains lower computational cost and higher accuracy than prior gradient-based methods. To bridge the gap between gradient-based affinity and data-driven prediction of MTL gains, ETAP applies a **learned non-linear mapping**, which corrects the scale mismatch between affinity scores and MTL gains. Finally, to incorporate data-driven predictions, ETAP applies a **learned residual predictor**, which can model higher-order, group-dependent interactions.
>
> Thus, ETAP is not merely an engineering combination of existing tools, but a novel framework that (i) retains the low-variance, model-aware information encoded in gradient-based affinity, (ii) corrects systematic bias through learned transformations, and (iii) captures higher-order interactions through residual modeling. These advantages lead to the improved prediction accuracy and sample efficiency demonstrated in our experiments.

---

> > ### Author Response · Authors · 2025-11-21
> >
> > ## W2: Scalability
> > We agree with the reviewer that scalability is an important concern. **ETAP is substantially more scalable than prior approaches for calculating affinity scores** in terms of computational complexity, which we clarify here.
> >
> > To compute pairwise affinity scores, ETAP requires training one MTL model with all the tasks. The computational cost of training this MTL model scales linearly ($O(n)$) with the number of tasks, so it does not pose a challenge for scalability. Note that many existing approaches also require training an MTL model with all the tasks (e.g., TAG from Fifty et al. (2021) and GRAD-TAE from Li et al. (2024)).
> >
> > ETAP also requires $O(n^2)$ extra operations during the training of this MTL model to compute pairwise affinity scores for the ${n \choose 2}$ pairs of tasks. However, these operations are **simple dot products of already-computed gradient vectors**, which require no additional forward or backward passes. Therefore, the computational cost of these operations is very low compared to the cost of training and does not pose a challenge for scalability in practice. This is in contrast to TAG, a similar approach for computing pairwise affinity scores, which requires $O(n^2)$ extra forward passes in each step; these extra forward passes incur a substantial computational burden.
> >
> > Beyond computational complexity, we would like to highlight that ETAP outperforms state-of-the-art data-driven methods, such as MTGNet (Song et al. 2022) and Linear Surrogate (Li et al. 2023), in experimental evaluation: ETAP can attain the same predictive accuracy at a lower computational cost (i.e., requiring fewer training groups $\mathcal{G}_{\text{train}}$); see, e.g., Figure 2.
> >
> > Finally, we would also like to highlight that prior work on predicting MTL gains used similar experimental settings with similar numbers of tasks (e.g., TAG and MTGNet).

---

> > > ### Author Response · Authors · 2025-11-21
> > >
> > > ## W3: Clarification on Branch-and-Bound Algorithm
> > > We thank the reviewer for bringing this concern to our attention. Our work indeed focuses on predicting MTL gains, providing more accurate and efficient predictions than existing approaches; our predictor may be integrated as an objective into a wide range of existing grouping algorithms (e.g., branch-and-bound algorithm from Standley et al. (2020) and Fifty et al. (2021) or beam-search algorithm from Song et al. (2022)). We added the following clarification to the end of the introduction (see right after “Contributions” in the revised PDF): *“Our MTL-gain predictor may be integrated as an objective into a wide range of search algorithms to find optimal task groups for MTL (e.g., it can be integrated into branch-and-bound algorithms from Standley et al. (2020) and Fifty et al. (2021) or beam-search algorithm from Song et al. (2022)). In our experimental evaluation, we use the efficient branch-and-bound search algorithm from Standley et al. (2020) to find task groups with our proposed predictor and with baseline predictors.”*

---

### Official Review · Reviewer_LwS8 · 2025-10-31

**Soundness:** 3
**Presentation:** 3
**Contribution:** 3
**Rating:** 6
**Confidence:** 4

**Summary:**

Summary

This paper addresses a central challenge in multi-task learning (MTL): identifying groups of tasks that mutually improve each other’s performance when trained together. Existing methods typically fall into two categories — white-box and black-box approaches.

The paper introduces a hybrid method that integrates both. The proposed two-step framework first estimates pairwise task affinities by training a single MTL model on all tasks within a group, inferring affinities for all possible combinations. In the second step, a non-linear mapping from task affinity to performance gain is constructed and refined using residual predictions.

The method is evaluated against several state-of-the-art task grouping algorithms, with ablation studies highlighting the contribution of each component. Results demonstrate the benefits of applying non-linear mappings and residual predictions. Overall, the paper presents a robust and well-conceived methodology that successfully combines the strengths of existing approaches.

**Strengths:**

Strengths

Tackles an important challenge in the MTL literature — the high computational overhead of existing task-grouping algorithms. Through comprehensive ablation studies, the paper provides valuable insights into the role of gradient similarities (e.g., via comparison of affine vs. non-linear mappings).

Design choices (e.g., use of B-splines and regression techniques) are justified through ablation analyses and comparative experiments.

The paper is clearly structured and effectively relates its contributions to prior work, ensuring coherence and contextual grounding.

The experimental evaluation goes beyond measuring performance gains, also analyzing the correlation between predicted affinities and ground-truth transfer gains.

**Weaknesses:**

Weaknesses

While computational efficiency is claimed as a key advantage, it would be valuable to include results for the complete approach (including hyperparameter tuning in the second stage) or to explicitly state that the additional cost is negligible. Comparing against an additional data-driven baseline would strengthen the evaluation.

The performance gains over naive MTL are relatively modest; a discussion of their practical significance would help contextualize their value.

Some design choices lack clear theoretical justification (e.g., why averaging gradient similarities yields effective affinity estimates, or why B-splines are particularly suitable).

The proposed method introduces several additional hyperparameters (for the B-spline expansion, regression method, and residual prediction), which may complicate tuning and reproducibility.

**Questions:**

Questions for the Authors

What is the rationale for time-averaging the cosine similarity over the
K
K training steps in Equation (6)? How stable is this measure across training, and how much variability is typically observed?

Since TAG’s affinity scores are on a different scale than observed gains, how does your method correct for this discrepancy?

You mention that GRAD-TAE performs well for groups — could you include these results in Table 3 for completeness?

Could you provide additional details on the computational requirements of the different methods, particularly in terms of the number of backward passes and the extent of hyperparameter tuning involved?

Recommendation

Although some design choices could be better motivated and the performance gains over naive MTL are modest, the paper provides valuable insights into the problem of task grouping. The authors conduct detailed and well-structured experiments, including informative ablation studies that clarify the role of different components within the proposed framework.

Additional Feedback

Figure 1 is somewhat difficult to interpret, as it presents too many elements at once. Consider splitting it into two complementary figures: one conceptual illustration to convey the overall idea, and another outlining the algorithmic steps in more detail.

Section 4 could be improved for clarity and consistency. Ensure that all discussed results appear in the corresponding tables or figures, and avoid repetition of statements like “TAG achieves higher correlation but incurs significant computational overhead.”

The term “training cost” could be replaced with “computational cost” for improved clarity.

In Section 3.1.3, explicitly explain the difference with TAG and refer to the appendix where both affinity measures are compared.

The discussion of limitations could be expanded. As shown in the appendix, the method performs comparably to naive MTL in some cases, indicating room for improvement and further exploration in future work.

**Details Of Ethics Concerns:**

No ethics concerns.

---

> ### Author Response · Authors · 2025-11-21
>
> ## Q1: Rationale for Time Averaging in Equation (6)
> The reviewer raises a very interesting question. We measure affinity at every training step because the **similarity between task-specific gradients varies naturally during training**; especially in early epochs, gradients fluctuate significantly. A single measurement (e.g., only the first step, last step, or any step in between) captures only a snapshot and provides a *high-variance, unreliable estimate of task affinity*. Averaging affinity scores over all $K$ training steps provides a **low-variance, robust estimate that captures task interactions throughout the entire training process**. In other words, time averaging the affinity scores is necessary for stable estimation because gradient similarities vary significantly during training.
>
> To answer the question regarding the stability of our time-averaged gradient similarities (i.e., our affinity score $\overline{z}_{{t_i} \rightarrow {t_j}}$), we reference Table 2, which shows the correlation between our affinity score (ETAP) and ground-truth MTL gains. This table presents aggregated results from a number of independent MTL training runs, reporting their mean and standard deviation (i.e., *mean* $\pm$ *std. dev.*). The results demonstrate that our affinity scores (ETAP) are **very stable between different training runs** since the standard deviation is very low compared to the mean.
>
> We added a clarification regarding this to the paper (see new paragraph after Equation (6) in the revised PDF).
>
> ### Variance of Per-Step Affinity Score During Training
>
> To illustrate how much the per-step affinity score $z^k_{{t_i} \rightarrow {t_j}}$ fluctuates during training, we plotted the score between task “5_o_Clock_Shadow” and each other tasks of the CelebA dataset over the course of a single MTL training run. Since we cannot post figures here, we added these plots as a figure in the paper appendix (see Figure 7 in Appendix F), which shows that similarities vary significantly during training.
>
> Here, we present summary statistics of the per-step affinity scores for two pairs of tasks from each dataset, measured over the entire course of a training:
> * mean score over all training steps,
> * standard deviation of score over all training steps (measure of variability),
> * coefficient of variation (std. dev. / mean),
> * range (minimum and maximum score over all training steps).
>
>
> These statistics illustrate that per-step affinity scores vary significantly during training, motivating the need for time averaging.
>
> | Dataset  | Task Pair        | Mean Score | Std. Dev. | Coeff. of Variation | Min.   | Max.  |
> | -------- | ---------------- | --------------- | --------- | ------------------- | ----- | ---- |
> | **CelebA**   |  A → B | 0.20            | 0.08      | 0.38                | 0.03 | 0.40 |
> |   | B→ C | 0.73            | 0.31      | 0.43               | 0.26 | 1.47 |
> | **ETTm1** | A→B | 0.71           | 2.14      | 3.02                | 0.11 | 9.00 |
> | | B→C | 0.16           | 0.56      | 3.47                | –0.02 | 2.34 |
> | **Chemical** | A→B | -0.01           | 0.01      | -0.46                | –0.02 | 0.001 |
> | | B→C | 0.039           | 0.01      | 0.24                | 0.01 | 0.041 |
> | **Ridership** | A→B | -3.13          | 3.76      | -1. 20               | –15.75 | -0.38 |
> | | A→C | -1.05          | 1.50      | -1. 43               | –4.95 | 0.49 |
>
> ### Benefit of Time Averaging
>
> To illustrate the benefit of time averaging, we computed affinity scores at three specific training steps (first step, middle step, and last steps), and we compared them to time-averaged scores (i.e., our affinity score $\overline{z}_{{t_i} \rightarrow {t_j}}$) based on their correlation to ground-truth MTL gains (higher is better). We present the results in the table below, which shows that time-averaged scores are more robust.
>
> | Dataset  | Step | Score Correlation ($\mu \pm \sigma$) |
> |----------|----------------|---------------------------|
> | **CelebA** | First   | -0.03 ± 0.00            |
> |          | Middle   | 0.34 ± 0.00             |
> |          | Last     | 0.29 ± 0.00             |
> |          | Time-averaged     | 0.32 ± 0.00            |
> | **ETTm1** | First   | 0.54 ± 0.00             |
> |             | Middle  | 0.17 ± 0.00             |
> |             | Last    | 0.08 ± 0.00             |
> |             | Time-averaged     | 0.47 ± 0.00             |
> | **Chemical** | First   | 0.08 ± 0.01             |
> |             | Middle  | 0.39 ± 0.01             |
> |             | Last    | 0.38 ± 0.02             |
> |             | Time-averaged     | 0.40 ± 0.03             |
> | **Ridership** | First   | 0.39 ± 0.02             |
> |             | Middle  | 0.32 ± 0.06             |
> |             | Last    | 0.29 ± 0.11             |
> |             | Time-averaged     | 0.36 ± 0.03             |

---

> > ### Author Response · Authors · 2025-11-21
> >
> > ## Q2: Discrepancy Between the Scales of Affinity Scores and MTL Gains
> > Since the affinity scores output by TAG and the observed MTL gains are on different numerical scales, our experimental evaluation (Tables 2 and 3) considers the **correlation** between them. This enables us to make a fair comparison between different methods, such as ETAP and TAG, since correlation is scale independent. We added a clarification regarding this to the paper (see paragraph below Table 2 in the revised PDF).
> >
> > We would also like to highlight that our method, ETAP, also calculates gradient-based affinity scores $\overline{z}$ that are on a different scale than MTL gains; however, **ETAP translates these affinity scores to MTL gains via a learned non-linear mapping**. Through this translation, ETAP bridges the gap between gradient-based and data-driven approaches for predicting MTL gains, which addresses some of the fundamental limitations of these two paradigms. On the one hand, prior gradient-based affinity approaches (e.g., TAG) treat gradient alignment as the final predictor, which suffers from (i) a scale mismatch between affinity scores and MTL gains and (ii) high prediction bias due to an implicit linearity assumption. On the other hand, prior data-driven methods (e.g., MTGNet and Linear Surrogate) treat MTL as a black box, relying only on measured MTL gains to make predictions, which suffers from (i) high prediction variance due to a lack of informative priors and (ii) poor, brittle performance when the number of training groups is low.
> >
> > To bridge the gap between gradient-based affinity scores (i.e., white-box approaches) and data-driven prediction (i.e., black-box approaches), ETAP first applies a learned non-linear mapping, which corrects the scale mismatch between affinity scores and MTL gains. Second, to incorporate data-driven predictions, ETAP applies a learned residual predictor, which can model higher-order, group-dependent interactions. By unifying gradient-based and data-driven approaches, ETAP (i) retains the low-variance, model-aware information encoded in gradient-based affinity, (ii) corrects systematic bias through learned transformations, and (iii) captures higher-order interactions through residual modeling. These advantages lead to the improved prediction accuracy and sample efficiency demonstrated in our experiments.

---

> > > ### Author Response · Authors · 2025-11-21
> > >
> > > ## Q3: GRAD-TAE for Group Prediction in Table 3
> > > We would like to clarify that while GRAD-TAE can estimate MTL gains for groups, these estimates are computationally very expensive. Due to this high computational cost, GRAD-TAE estimates cannot be incorporated directly into search algorithms for group selection. For these reasons, GRAD-TAE is not directly comparable to methods for MTL gain prediction in Table 3. Following the reviewer’s suggestion, we did perform additional experiments with GRAD-TAE, where we translated GRAD-TAE estimates into pairwise affinities using the GRAD-TAE framework. We included these experimental results in Table 4 (see the revised PDF), and we will also discuss them below.
> > >
> > > First, let us explain the reason behind the high computational cost of GRAD-TAE. GRAD-TAE estimates MTL gains for a group of tasks by (i) calculating task-specific loss gradients on a pre-trained baseline model, (ii) performing linearized fine-tuning for this group by weighing the gradients using logistic regression, (ii) and measuring test loss for each task in the group on this estimated fine-tuned model. The computational cost of these steps is high; for example, 24 ± 4.8 seconds per group on CelebA and 35 ± 1.8 seconds per group on ETTm1. In contrast, ETAP can predict MTL gains for a group in 0.00002 seconds on CelebA and in 0.00008 seconds on ETTm1 (once it has been trained); MTGNet and TAG also have low computational cost per prediction. **Since GRAD-TAE incurs high computational cost for estimating the MTL gains of each group, these estimates cannot be incorporated directly into a search algorithm that may need to consider millions of possible groups and their MTL gains.**
> > >
> > > Instead, GRAD-TAE estimates MTL gains for only a *limited number of randomly sampled groups* and derives *pairwise affinity estimates from these samples*. GRAD-TAE then uses these pairwise affinity estimates as input for group selection.
> > >
> > > ### Comparison with GRAD-TAE in Terms of Group Selection
> > >
> > > To provide a fair comparison between different prediction methods, we used the pairwise affinity estimates of GRAD-TAE with the same branch-and-bound search algorithm as the other prediction methods (TAG, MTGNet, and ETAP). The table below shows the results of grouping tasks based on various prediction methods, including GRAD-TAE (Table 4 in our revised PDF). These results confirm that ETAP achieves competitive or superior grouping quality across the datasets, while incurring low computational cost.
> > >
> > > #### Table 4 (revised): MTL performance (measured as total loss (± std); lower is better) with tasks grouped based on various prediction methods, including GRAD-TAE.
> > >
> > > | Dataset     | Groups | TAG  | GRAD-TAE  | MTGNet ($\| \mathcal{G}_{\text{train}}\| $) | ETAP  ($\| \mathcal{G}_{\text{train}}\| $)  | Optimal |
> > > |----------------|--------|------|----------|------------------------|------------------------|---------|
> > > | **CelebA**     | 2      | 49.67 ± 0.00 | 50.78 ± 0.59 | 50.62 ± 1.20 | 49.92 ± 0.23 | 49.27 |
> > > | ($\|\mathcal{G}_{\text{train}}\|$ = 10) | 3   | 50.22 ± 0.00 | 50.78 ± 1.93 | 50.31 ± 0.63 | 49.61 ± 0.30 | 48.63 |
> > > |                | 4      | 49.94 ± 0.00 | 52.83 ± 1.70 | 50.27 ± 0.60 | 49.41 ± 0.25 | 48.38 |
> > > | **ETTm1**      | 2      | 4.08 ± 0.09  | 4.08 ± 0.06 | 4.04 ± 0.06  | 4.02 ± 0.05  | 3.98 |
> > > | ($\|\mathcal{G}_{\text{train}}\|$= 10) | 3   | 3.96 ± 0.03  | 4.04 ± 0.06 | 3.96 ± 0.03  | 3.93 ± 0.07  | 3.83 |
> > > |                | 4      | 3.90 ± 0.02  | 4.01 ± 0.04 | 3.92 ± 0.06  | 3.89 ± 0.09  | 3.82 |
> > > | **Chemical**   | 2      | 4.69 ± 0.11  | 4.80 ± 0.14 | 4.79 ± 0.31  | 4.67 ± 0.12  | 4.56 |
> > > | ($\|\mathcal{G}_{\text{train}}\|$= 10) | 3   | 4.80 ± 0.06  | 4.83 ± 0.05 | 4.89 ± 0.15  | 4.74 ± 0.07  | 4.52 |
> > > |                | 4      | 4.95 ± 0.08  | 4.96 ± 0.10 | 4.94 ± 0.09  | 4.83 ± 0.10  | 4.67 |
> > > | **Ridership**  | 2      | 17.50 ± 0.00 | 18.48 ± 0.80 | 17.86 ± 0.43 | 17.77 ± 0.26 | 17.03 |
> > > | ($\|\mathcal{G}_{\text{train}}\|$ = 10) | 3   | 18.31 ± 0.00 | 17.94 ± 0.47 | 18.12 ± 0.30 | 17.83 ± 0.26 | 16.90 |
> > > |                | 4      | 18.25 ± 0.00 | 18.45 ± 0.36 | 18.06 ± 0.42 | 17.59 ± 0.19 | 16.79 |

---

> > > > ### Author Response · Authors · 2025-11-21
> > > >
> > > > ## Q4: Details on Computational Requirements and Hyperparameter Tuning
> > > > The computational cost of ETAP is dominated by the cost of training MTL models, which are used for (i) estimating pairwise affinity scores and (ii) obtaining ground-truth MTL gains for a small set of training groups $\mathcal{G}_{\text{train}}$. The (i) calculation of cosine similarities between gradients and the (ii) training of regression models incur **negligible computational overhead**. The computational cost of ETAP can be broken down into the following steps:
> > > >
> > > > 1. *Measuring task-affinity scores (Section 3.1):* This step requires training one MTL model with all the tasks, and calculating similarities between task-specific gradients during the training (Equation (5)). Since the similarity calculation is a *simple dot-product operation over already-computed gradient vectors*, it requires no additional forward or backward passes, so its computational cost is very low (unlike TAG, which requires additional forward passes). Note that training an MTL model with all the tasks is scalable in the sense that its complexity is $O(n)$ (i.e., linear in the number of tasks), since it requires a forward and backward pass for each task in each epoch (i.e., standard gradient descent).
> > > >
> > > > 2. *Measuring ground-truth MTL gain for training groups $\mathcal{G}_{\text{train}}$:* This step requires training an MTL model for each training group. This cost is shared by all data-driven methods (e.g., MTGNet and Linear Surrogate). Crucially, ETAP is more efficient than existing data-driven methods since it can accurately predict MTL gains based on very small sets of training groups (e.g., $|\mathcal{G}_{\text{train}}| = 5$ or $10$), while other methods require much larger training sets (see Figure 2 in the paper).
> > > >
> > > > 3. *Training regression models (Sections 3.2):* This step requires training regression models with B-spline bases for mapping affinity scores (Section 3.2.1) and with multi-hot encoding for residual prediction (Section 3.2.2). These models are **several orders of magnitude smaller** than any MTL model, so the computational cost of their training is negligible compared to the cost of training an MTL model. As a result, the **cost of hyperparameter tuning is also negligible**, even with extensive cross validation.
> > > >
> > > > ### Computational Cost of Hyperparameter Tuning
> > > >
> > > > We search for optimal spline degree, number of knots, and ridge-regularization strength using cross validation. We report the running time of these hyperparameter searches in the table below (with $|\mathcal{G}_{\text{train}}|=5$):
> > > >
> > > > | Dataset | B-Spline Tuning (Degree and Knots) | Regularization Tuning|
> > > > |----------|---------------------|--------------------|
> > > > | CelebA| 15.63 s                |1.41 s                 |
> > > > |ETTm1| 13.77 s                 | 1.30 s                |
> > > > |Chemical| 15.82 s              |1.64 s                 |
> > > > |Ridership| 15.23 s              |1.23 s                 |
> > > >
> > > > The running time of training an MTL model depends on the size of the group. We report the average running times for MTL training in the table below:
> > > >
> > > > | Dataset | MTL Training |
> > > > |--------|--------------|
> > > > |CelebA| 52.56 ± 18.93 minutes |
> > > > |Chemical| 1.2 ± 0.2 minutes |
> > > > |ETTm1| 16 ± 4 minutes |
> > > > |Ridership| 9 ± 1.1 minutes |
> > > >
> > > > Across all benchmarks, ETAP’s hyperparameter tuning requires only **15–18 seconds in total**. For the CelebA, ETTm1, and Ridership datasets, this corresponds to less than 3% of the cost of training a single MTL model. Even for the Chemical dataset, it is a small fraction. Therefore, in all cases, the computational cost of hyperparameter tuning is negligible compared to the cost of training MTL models.

---

> > > > > ### Author Response · Authors · 2025-11-21
> > > > >
> > > > > Following the reviewer’s suggestion, we replaced the term “training cost” with "computational cost” for improved clarity throughout the paper (see revised PDF).
> > > > >
> > > > > ## Comparison with Additional Data-Driven Baseline
> > > > > To strengthen the evaluation, we conducted additional experiments using the recent data-driven method of **Ayman et al. (2023), “Task Grouping for Automated Multi-Task Machine Learning via Task Affinity Prediction”**, and included the results in the revised PDF.
> > > > >
> > > > > The method of Ayman et al. (2023) [2] was designed for **tabular datasets**, whereas ETAP can be applied to a wider range of datasets, including **computer vision**, **time-series**, and **tabular data**. Crucially, the method of [2] relies heavily on **features whose computation is expensive**. For example, one of its features, the *mean pairwise MTL gain for a group* requires training MTL models for *all* ${n \choose 2}$ task pairs. This is computationally very expensive and may be prohibitive for larger task sets. Nevertheless, since we have access to ground-truth MTL gains for all task pairs in our two tabular benchmarks (Chemical and Ridership), we implemented and evaluated the method of [2] on these benchmark datasets. For these experiments, we implemented the exact set of features and neural network specified in [2].
> > > > >
> > > > > Another major limitation of the method of [2] is that it predicts the **total MTL gain for all tasks in a group**, instead of predicting **MTL gain for each task in a group** as ETAP and most other methods do (e.g., TAG and MTGNet). For a fair comparison, we adapted ETAP to the evaluation setting of [2]: we used ETAP to predict MTL gain for each task in a group, and then we summed these gains up to obtain a predicted total MTL gain for the group. We present the results of these experiments in the table below, which we also included in our paper (Table 5 in the revised PDF).
> > > > >
> > > > > ### Table 5 (new): Correlation and R² between ground-truth and predicted total MTL gains for groups (higher values are better).
> > > > >
> > > > > | **Dataset**       | Training Groups | **Ayman et al. (2023) Correlation** | **ETAP Correlation** | **Ayman et al. (2023) R²** | **ETAP R²** |
> > > > > | ------------- | ---------- | ---------------------- | -------------------- | --------------------- | ------------------- |
> > > > > | **Chemical**  | 5          | 0.33 ± 0.69           | **0.74 ± 0.04**      | –0.84 ± 2.12        | **0.15 ± 0.13**    |
> > > > > |               | 10         | 0.76 ± 0.05           | **0.75 ± 0.05**      | 0.48 ± 0.12         | **0.36 ± 0.05**    |
> > > > > |               | 20         | 0.78 ± 0.06           | **0.77 ± 0.04**      | 0.56 ± 0.12         | **0.39 ± 0.19**    |
> > > > > | **Ridership** | 5          | –0.01 ± 0.13          | **0.44 ± 0.06**      | –0.18 ± 0.67        | **0.16 ± 0.08**    |
> > > > > |               | 10         | 0.05 ± 0.14           | **0.42 ± 0.07**      | –0.81 ± 0.57        | **0.12 ± 0.09**    |
> > > > > |               | 20         | 0.13 ± 0.08           | **0.46 ± 0.03**      | –0.25 ± 0.13        | **0.18 ± 0.05**    |
> > > > >
> > > > > As discussed above, the computational cost of the method of [2] includes the substantial cost of training MTL models for all ${n \choose 2}$ pairs of tasks. Further, the method also requires measuring ground-truth MTL gains for a set of training groups $G_{\text{train}}$, which incurs the cost of training $|G_{\text{train}}|$ MTL models for these groups. In contrast, ETAP achieves better accuracy at a lower cost: our method requires training only one MTL model to compute affinity scores (with an additional ${n \choose 2}$ dot-product operations over already-computed gradient vectors, which incurs very low overhead) and measuring ground-truth MTL gains for a set of training groups $G_{\text{train}}$. Since the number of training groups in most practical applications can be at least an order of magnitude lower than ${n \choose 2}$, **ETAP incurs a significantly lower overall computational cost**.

---

### Official Review · Reviewer_Y2PA · 2025-11-01

**Soundness:** 2
**Presentation:** 3
**Contribution:** 2
**Rating:** 4
**Confidence:** 3

**Summary:**

Proposed ETAP builds a scalable predictor by computing a gradient-alignment affinity score for pairs and groups in shared parameters, then refining it with learned nonlinear transformations and residual corrections. Across benchmarks, ETAP improves MTL gain prediction and enables more effective task grouping, outperforming used baselines.

**Strengths:**

It combines gradient-based affinity with learned non-linear relationship modeling to efficiently and accurately capture task relationships. And it includes thorough component-wise ablations that clarify each contribution and improve interpretability.

**Weaknesses:**

Dividing learning tasks into groups based on similarity is a long-standing area [1]. The paper introduces new measures of task affinity for MTL, but I am not fully convinced that the proposed methods are superior to prior work. The baselines used are relatively dated, and a comparison of computational cost and predictive performance with stronger recent baselines, such as [2], would strengthen the claims. Efficient group-wise tracking of task affinity is also not new, as [3] tracks inter-task affinity in a group-wise manner during multi-task optimization. Finally, I am not convinced that the proposed methods clearly improve over other inter-task affinity tracking approaches, including [4].

[1] Which tasks should be learned together in multi-task learning?

[2] Task Grouping for Automated Multi-Task Machine Learning via Task Affinity Prediction

[3] Selective Task Group Updates for Multi-Task Optimization

[4] Scalable Multitask Learning Using Gradient-based Estimation of Task Affinity

**Questions:**

It would help to clarify the concrete differences from prior work, especially how your methods distinctively improve on gradient-based affinity and group-wise tracking approaches. Practical guidance on when to prefer your method over more recent baselines would make the contribution clearer.

---

> ### Author Response · Authors · 2025-11-21
>
> We thank the reviewer for the valuable feedback and suggestions. To answer the reviewer’s question, we would like to clarify that the main novelty of ETAP lies in **bridging the gap between gradient-based and data-driven approaches for predicting MTL gains**. By introducing this novel paradigm, ETAP addresses the fundamental limitations of both gradient-based and data-driven methods. On the one hand, prior gradient-based affinity methods (e.g., TAG and suggested baseline [3]) treat gradient alignment as the final predictor, which suffers from (i) a scale mismatch between affinity scores and MTL gains and (ii) high prediction bias due to an implicit linearity assumption. On the other hand, prior data-driven methods (e.g., MTGNet and Linear Surrogate) treat MTL as a black box, relying only on measured MTL gains to make predictions, which suffers from (i) high prediction variance due to a lack of informative priors and (ii) poor, brittle performance when the number of training groups is low.
>
> ETAP uniquely bridges these two perspectives by introducing a principled two-stage prediction ensemble that combines gradient-based affinity (i.e., white-box approach) with data-driven prediction (i.e., black-box approach). Adopting this paradigm requires a number of innovations: ETAP first computes **low-cost gradient-based affinity scores** using a novel method, which attains lower computational cost and higher accuracy than prior gradient-based methods. To bridge the gap between gradient-based affinity and data-driven prediction of MTL gains, ETAP applies a **learned non-linear mapping**, which corrects the scale mismatch between affinity scores and MTL gains. Finally, to incorporate data-driven predictions, ETAP applies a **learned residual predictor**, which can model higher-order, group-dependent interactions.
>
> By unifying gradient-based and data-driven approaches, ETAP (i) retains the low-variance, model-aware information encoded in gradient-based affinity, (ii) corrects systematic bias through learned transformations, and (iii) captures higher-order interactions through residual modeling. These advantages lead to the improved prediction accuracy and sample efficiency demonstrated in our experiments. Our paper includes comparisons with several relevant and recent baselines, including TAG (Fifty et al., 2021), MTGNet (Song et al., 2022), Grad-TAE (Li et al., 2024), Linear Surrogate (Li et al., 2024), and PCGrad (Yu et al., 2020). We agree with the reviewer that including additional baselines and comparisons will strengthen our results, further clarifying the advantages of ETAP in terms of computational cost and prediction accuracy. To this end, **we conducted additional experiments with the suggested baseline [2]** (Ayman et al., 2023. Task Grouping for Automated Multi-Task Machine Learning via Task Affinity Prediction) and **with GRAD-TAE [4]** (Li et al., 2024. Scalable Multitask Learning Using Gradient-based Estimation of Task Affinity).

---

> > ### Author Response · Authors · 2025-11-21
> >
> > ## Comparison with Baseline Method of Ayman et al. (2023) [2]
> > The method of Ayman et al. (2023) [2] was designed for **tabular datasets**, whereas ETAP can be applied to a wider range of datasets, including **computer vision**, **time-series**, and **tabular data**. Crucially, the method of [2] relies heavily on **features whose computation is expensive**. For example, one of its features, the *mean pairwise MTL gain for a group* requires training MTL models for *all* ${n \choose 2}$ task pairs. This is computationally very expensive and may be prohibitive for larger task sets. Nevertheless, since we have access to ground-truth MTL gains for all task pairs in our two tabular benchmarks (Chemical and Ridership), we implemented and evaluated the method of [2] on these benchmark datasets. For these experiments, we implemented the exact set of features and neural network specified in [2].
> >
> > Another major limitation of the method of [2] is that it predicts the **total MTL gain for all tasks in a group**, instead of predicting **MTL gain for each task in a group** as ETAP and most other methods do (e.g., TAG and MTGNet). For a fair comparison, we adapted ETAP to the evaluation setting of [2]: we used ETAP to predict MTL gain for each task in a group, and then we summed these gains up to obtain a predicted total MTL gain for the group. We present the results of these experiments in the table below, which we also included in our paper (Table 5 in the revised PDF).
> >
> > ### Table 5 (new): Correlation and R² between ground-truth and predicted total MTL gains for groups (higher values are better).
> >
> > | **Dataset**       | Training Groups | **Ayman et al. (2023) Correlation** | **ETAP Correlation** | **Ayman et al. (2023) R²** | **ETAP R²** |
> > | ------------- | ---------- | ---------------------- | -------------------- | --------------------- | ------------------- |
> > | **Chemical**  | 5          | 0.33 ± 0.69           | **0.74 ± 0.04**      | –0.84 ± 2.12        | **0.15 ± 0.13**    |
> > |               | 10         | 0.76 ± 0.05           | **0.75 ± 0.05**      | 0.48 ± 0.12         | **0.36 ± 0.05**    |
> > |               | 20         | 0.78 ± 0.06           | **0.77 ± 0.04**      | 0.56 ± 0.12         | **0.39 ± 0.19**    |
> > | **Ridership** | 5          | –0.01 ± 0.13          | **0.44 ± 0.06**      | –0.18 ± 0.67        | **0.16 ± 0.08**    |
> > |               | 10         | 0.05 ± 0.14           | **0.42 ± 0.07**      | –0.81 ± 0.57        | **0.12 ± 0.09**    |
> > |               | 20         | 0.13 ± 0.08           | **0.46 ± 0.03**      | –0.25 ± 0.13        | **0.18 ± 0.05**    |
> >
> > As discussed above, the computational cost of the method of [2] includes the substantial cost of training MTL models for all ${n \choose 2}$ pairs of tasks. Further, the method also requires measuring ground-truth MTL gains for a set of training groups $G_{\text{train}}$, which incurs the cost of training $|G_{\text{train}}|$ MTL models for these groups. In contrast, ETAP achieves better accuracy at a lower cost: our method requires training only one MTL model to compute affinity scores (with an additional ${n \choose 2}$ dot-product operations over already-computed gradient vectors, which incurs very low overhead) and measuring ground-truth MTL gains for a set of training groups $G_{\text{train}}$. Since the number of training groups in most practical applications can be at least an order of magnitude lower than ${n \choose 2}$, **ETAP incurs a significantly lower overall computational cost**.

---

> > > ### Author Response · Authors · 2025-11-21
> > >
> > > ## Comparison with GRAD-TAE [4] (Li et al., 2024) in Terms of Group Selection
> > > To address the reviewer’s concern about whether ETAP clearly improves over other inter-task affinity tracking approaches, specifically GRAD-TAE [4], we provide additional experimental results on group selection with GRAD-TAE.
> > >
> > > GRAD-TAE estimates MTL gains for a group of tasks by (i) calculating task-specific loss gradients on a pre-trained baseline model, (ii) performing linearized fine-tuning for this group by weighing the gradients using logistic regression, (ii) and measuring test loss for each task in the group on this estimated fine-tuned model. These group-level estimates are relatively expensive to compute (e.g., GRAD-TAE takes 24 ± 4.8 seconds per group estimate on CelebA and 35 ± 1.8 seconds per group estimate on ETTm1; while ETAP takes 0.00002 and 0.00008 seconds per group prediction, respectively). Due to this high computational cost, GRAD-TAE estimates MTL gains for only a *limited number of randomly sampled groups* and derives *pairwise affinity estimates from these samples*. These pairwise affinity estimates are then used as input for group selection.
> > >
> > > To provide a fair comparison between different prediction methods, **we used the pairwise affinity estimates of GRAD-TAE with the same branch-and-bound search algorithm as the other prediction methods** (TAG, MTGNet, and ETAP). The table below shows the results of grouping tasks with various prediction methods, including GRAD-TAE (Table 4 in our revised PDF). These results confirm that ETAP achieves competitive or superior grouping quality across the datasets, while incurring low computational cost.
> > >
> > > ### Table 4 (revised): MTL performance (measured as total loss (± std); lower is better) with tasks grouped based on various prediction methods, including GRAD-TAE.
> > >
> > > | Dataset     | Groups | TAG  | GRAD-TAE  | MTGNet ($\| \mathcal{G}_{\text{train}}\| $) | ETAP  ($\| \mathcal{G}_{\text{train}}\| $)  | Optimal |
> > > |----------------|--------|------|----------|------------------------|------------------------|---------|
> > > | **CelebA**     | 2      | 49.67 ± 0.00 | 50.78 ± 0.59 | 50.62 ± 1.20 | 49.92 ± 0.23 | 49.27 |
> > > | ($\|\mathcal{G}_{\text{train}}\|$ = 10) | 3   | 50.22 ± 0.00 | 50.78 ± 1.93 | 50.31 ± 0.63 | 49.61 ± 0.30 | 48.63 |
> > > |                | 4      | 49.94 ± 0.00 | 52.83 ± 1.70 | 50.27 ± 0.60 | 49.41 ± 0.25 | 48.38 |
> > > | **ETTm1**      | 2      | 4.08 ± 0.09  | 4.08 ± 0.06 | 4.04 ± 0.06  | 4.02 ± 0.05  | 3.98 |
> > > | ($\|\mathcal{G}_{\text{train}}\|$= 10) | 3   | 3.96 ± 0.03  | 4.04 ± 0.06 | 3.96 ± 0.03  | 3.93 ± 0.07  | 3.83 |
> > > |                | 4      | 3.90 ± 0.02  | 4.01 ± 0.04 | 3.92 ± 0.06  | 3.89 ± 0.09  | 3.82 |
> > > | **Chemical**   | 2      | 4.69 ± 0.11  | 4.80 ± 0.14 | 4.79 ± 0.31  | 4.67 ± 0.12  | 4.56 |
> > > | ($\|\mathcal{G}_{\text{train}}\|$= 10) | 3   | 4.80 ± 0.06  | 4.83 ± 0.05 | 4.89 ± 0.15  | 4.74 ± 0.07  | 4.52 |
> > > |                | 4      | 4.95 ± 0.08  | 4.96 ± 0.10 | 4.94 ± 0.09  | 4.83 ± 0.10  | 4.67 |
> > > | **Ridership**  | 2      | 17.50 ± 0.00 | 18.48 ± 0.80 | 17.86 ± 0.43 | 17.77 ± 0.26 | 17.03 |
> > > | ($\|\mathcal{G}_{\text{train}}\|$ = 10) | 3   | 18.31 ± 0.00 | 17.94 ± 0.47 | 18.12 ± 0.30 | 17.83 ± 0.26 | 16.90 |
> > > |                | 4      | 18.25 ± 0.00 | 18.45 ± 0.36 | 18.06 ± 0.42 | 17.59 ± 0.19 | 16.79 |

---

### Author Response · Authors · 2025-12-03
**Summary**

Dear AC,

Thank you for your time and effort in overseeing the review process under these extraordinary circumstances.

Our paper introduces ETAP, a scalable hybrid framework that integrates gradient-based affinity analysis with data-driven non-linear estimators to predict gains from multi-task learning, significantly outperforming baseline approaches in terms of prediction accuracy and computational efficiency. We are pleased to note that the majority of reviews (Reviewers `LwS8` and `66hX`) recommend acceptance. There is a consensus among all three reviewers regarding the paper’s primary strengths: all three reviewers recognize the **importance of the problem** for multi-task learning, praise ETAP’s **two-stage ensemble for integrating gradient-based and data-driven prediction paradigms**, acknowledge the **significant improvements over baseline approaches** in experiments, and commend the **comprehensive ablation studies**.

During the rebuttal period, we revised the paper to include additional experiments and clarifications that address the reviewers’ questions. We specifically addressed the concerns of Reviewer `Y2PA` (who gave the lowest rating but also the lowest confidence score) by including experiments with a new baseline approach [2] that was recommended by the reviewer and by extending the experiments with another baseline approach [4]. We also clarified the conceptual novelty of our paper: bridging the gap between white-box (i.e., gradient-based) and black-box (i.e., data-driven) approaches for predicting MTL gains. We believe that our revisions address all of the reviewers’ concerns (unfortunately, the reviewers did not have time to update their reviews before the abrupt end of the discussion period).
## Reviewer Y2PA
**Conceptual novelty:** We clarified that ETAP bridges the gap between gradient-based and data-driven approaches, addressing the fundamental limitations of these two paradigms while retaining their advantages. This requires a number of innovations, such as addressing the scale mismatch between affinity scores and MTL gains through a learned non-linear mapping.
**Comparison with baseline method [2]:** We performed experiments with the baseline method [2] recommended by the reviewer. The results show that ETAP either matches or significantly outperforms this baseline (Table 5). We also added a discussion of the limitations of this baseline (applicable only to tabular data) and its high computational cost ($O(n^2)$ training runs) to the paper.
**Comparison with baseline method [4]:** To demonstrate clear improvement over this “inter-task affinity tracking approach” [4], we performed additional experiments with affinity-based group selection. The results show that ETAP leads to significantly better grouping than this baseline (GRAD-TAE in Table 4).
## Reviewer LwS8
**Rationale for time-averaging (Q1):** We clarified that the time averaging in Equation (6) is necessary since similarity varies significantly during training; the average captures overall similarity. We also performed a number of new experiments, demonstrating both the substantial fluctuation of per-step similarity and the robustness of time-averaged similarity (comment below and Figure 7).
**Difference between the scales of affinity scores and MTL gains (Q2):** We clarified that this is indeed a major challenge, which is not addressed by prior work (TAG), and that ETAP addresses it through a learned non-linear mapping.
**GRAD-TAE for group prediction (Q3):** We clarified that GRAD-TAE cannot be applied directly to group prediction due to its computational cost; the GRAD-TAE framework derives pairwise affinities and uses those for task grouping. We performed additional experiments with GRAD-TAE following this approach, showing that ETAP leads to significantly better grouping (Table 4).
**Computational costs and hyperparameter tuning (Q4, W1, W4):** We provided runtime measurements and a detailed discussion of ETAP's computational complexity, clarifying that its overhead (including hyperparameter tuning) is negligible compared to the cost of MTL training.
**Additional data-driven baseline (W1):** The new baseline [2] recommended by Reviewer `Y2PA` is data-driven.
## Reviewer 66hX
**Clarification of novelty (W1):** We clarified novelty, similar to our first response to Reviewer `Y2PA`.
**Scalability (W2):** We provided a detailed discussion of computational costs, explaining that the $O(n^2)$ operations in ETAP are low-cost dot products (in contrast to the baseline TAG, which requires $O(n^2)$ additional forward passes). Training one MTL model across all tasks incurs $O(n)$ cost, which is required by many approaches (e.g., TAG and GRAD-TAE); hence, ETAP is scalable. We also highlighted that empirically, ETAP attains the same (or higher) predictive accuracy at a lower cost than baseline data-driven approaches.
**Clarification on branch-and-bound algorithm (W3):** We added a clarification following the reviewer’s suggestion.

---

### Meta-Review · Area_Chair_Rg13 · 2026-01-02

**Summary:**

This paper proposes an Ensemble Task Affinity Predictor to predict the performance gains of joint multi-task training, which addresses an important problem in the field of multi-task learning. The reviewers’ main concerns focus on the discussion and comparison with relevant baselines, the analysis of method complexity, and the motivation behind the design of certain steps. The authors are encouraged to incorporate these revisions in the final version. In addition, further discussion of the practical application scenarios of affinity computation and improvements over naive multi-task learning in terms of performance gains is necessary.

**Reviewer Concerns:**

Reviewer LwS8 and Reviewer 66hX gave positive ratings. The main concerns raised by Reviewer Y2PA and Reviewer LwS8 were about comparisons and discussions with related methods, which the authors addressed through revisions. The authors also responded positively to Reviewer LwS8’s questions regarding the motivation behind the method design and Reviewer 66hX’s concerns about scalability and complexity under large-scale tasks. Overall, the reviewers’ concerns were well addressed during the rebuttal period.

**Reviewer Scores:**

Reviewer Y2PA’s main concern is the comparison with prior methods (such as Ayman et al. (2023) and GRAD-TAE (Li et al., 2024)). During the rebuttal, the authors added discussions and experimental comparisons between the proposed method and these approaches, which addresses the reviewer’s concerns.

Reviewer LwS8 gave a positive initial rating and suggested that the authors discuss the computational cost of the proposed method, the motivation behind certain design choices, and its practical value. During the rebuttal, the authors added comparisons of training time and discussed the design motivations of several core components.

Reviewer 66hX was supportive and suggested discussing the differences from the TAG baseline, as well as adding discussion on large-scale tasks. The authors added the corresponding discussions and complexity analysis.

---

### Decision · Program_Chairs · 2026-01-26

Accept (Poster)